# Epigenomic regulation of human T-cell leukemia virus by chromatin-insulator CTCF

Xiaogang Cheng[1⊙], Ancy Joseph[1⊙], Victor Castro[1], Alice Chen-Liaw[1], Zachary Skidmore[1], Takaharu Ueno[2], Jun-ichi Fujisawa[2], Daniel A. Rauch[1], Grant A. Challen[1], Michael P. Martinez[3,4], Patrick Green[3,4], Malachi Griffith[1], Jacqueline E. Payton[5], John R. Edwards[1,6], Lee Ratner[1,7]*

1 Department of Medicine, Washington University School of Medicine, St Louis, Missouri, United States of America, 2 Department of Microbiology, Kansai Medical University, Osaka, Japan, 3 Center for Retrovirus Research, The Ohio State University, Columbus, Ohio, United States of America, 4 Department of Veterinary Biosciences, The Ohio State University, Columbus, Ohio, United States of America, 5 Department of Pathology and Immunology, Washington University School of Medicine, St Louis, Missouri, United States of America, 6 Department of Phamacogenomics, Washington University School of Medicine, St Louis, Missouri, United States of America, 7 Department of Molecular Microbiology, Washington University School of Medicine, St Louis, Missouri, United States of America

⊙ These authors contributed equally to this work.
* lratner@wustl.edu

**Data Availability Statement:** All relevant data are within the manuscript, its Supporting Information files and ChIP-seq data is available at GEO (accession number: GSE172194).

## Abstract

Human T-cell leukemia virus type 1 (HTLV-1) is a retrovirus that causes an aggressive T-cell malignancy and a variety of inflammatory conditions. The integrated provirus includes a single binding site for the epigenomic insulator, CCCTC-binding protein (CTCF), but its function remains unclear. In the current study, a mutant virus was examined that eliminates the CTCF-binding site. The mutation did not disrupt the kinetics and levels of virus gene expression, or establishment of or reactivation from latency. However, the mutation disrupted the epigenetic barrier function, resulting in enhanced DNA CpG methylation downstream of the CTCF binding site on both strands of the integrated provirus and H3K4Me3, H3K36Me3, and H3K27Me3 chromatin modifications both up- and downstream of the site. A majority of clonal cell lines infected with wild type HTLV-1 exhibited increased plus strand gene expression with CTCF knockdown, while expression in mutant HTLV-1 clonal lines was unaffected. These findings indicate that CTCF binding regulates HTLV-1 gene expression, DNA and histone methylation in an integration site dependent fashion.

## Author summary

Human T-cell leukemia virus type 1 (HTLV-1) is a cause of leukemia and lymphoma as well as several inflammatory medical disorders. The virus integrates in the host cell DNA, and it has a single binding site for a protein designated CTCF. This protein is important in the regulation of many DNA viruses, as well as many properties of normal and malignant cells. In order to define the role of CTCF binding to HTLV, we analyzed a mutant virus lacking the binding site. We found that this mutation variably affected gene

**Funding:** This work was supported by Public Health Service grants to GAC (HL147978, DK124883 and DK102428), LR (AI26652, CA63417) and PG (CA100730). GAC is a scholar of the Leukemia and Lymphoma Society. The funders had no role in study design, data collection and analysis, decision to publish, or preparation of the manuscript.

**Competing interests:** The authors have declared that no competing interests exist.

expression, DNA and histone modification, suggesting a key role in regulation of virus replication in infected cells.

## Introduction

Human T-cell leukemia virus type-1 (HTLV-1) is a primate retrovirus which has infected about 10 million people worldwide [1]. Approximately 5% of infected individuals develop an aggressive malignancy of CD4$^+$ T-cells known as adult T-cell leukemia/lymphoma (ATL) or HTLV-1-associated myelopathy/tropical spastic paraparesis (HAM/TSP) after a prolonged period of clinical latency [2]. The remaining HTLV-1-infected individuals are at risk for development of other inflammatory disorders, or remain as asymptomatic carriers (AC) [3]. Despite advances in HTLV research, the precise mechanism for development of ATL and HAM/TSP remains incompletely understood.

HTLV-1 reverse-transcribes its 9 kb genomic RNA into complementary double-stranded DNA which is then integrated into the host DNA upon infection [4]. Thereafter the virus remains as a chromatinized provirus and it is replicated as a part of the host genome. High-throughput genome-wide sequencing of HTLV-1 integration sites from ACs, patients with HAM/TSP and patients with ATL, combined with HTLV-1 infected T-cells infected in culture, revealed that HTLV-1 integration strongly favors actively transcribed regions of the genome [5]. The HTLV-1 host carries between $10^4$ and $10^5$ infected T-cell clones, each clone carrying a single copy of the provirus in a unique genomic location [6]. A large number of HTLV-1-infected clones are established early in infection, after which persistent clonal proliferation maintains a stable hierarchy of HTLV-1-infected clones for the remainder of the host's life [7]. The viral regulatory proteins, Tax (transactivator protein) and HBZ (helix basic zipper protein), which are encoded respectively by the sense and antisense strands of the provirus, play indispensable roles in pathogenesis [8]. Tax is the primary oncogene of HTLV-1 [9]. Tax drives proviral transcription via transactivation of the 5′ long terminal repeat (LTR) and promotes cellular proliferation via dysregulation of multiple host genes. HBZ acts as a negative regulator of Tax-mediated host gene transcription and viral expression, and promotes cellular proliferation in both its protein and RNA forms and it also selectively inhibits activation of the p65-mediated classical NF-κB pathway [10–12].

The zinc finger protein, CCCTC-binding factor (CTCF), is a DNA binding factor capable of regulating not only the 3D genome organization, but also many key aspects of gene expression, including transcriptional activation and repression, RNA splicing, and enhancer/promoter insulation [13]. Several tumorigenic viruses, including Kaposi's sarcoma-associated herpesvirus, human papillomavirus, and Epstein Barr virus utilize CTCF to regulate viral gene expression [14]. Recently, CTCF was found to bind to the pX region of the HTLV-1 provirus, the portion of the provirus between *env* and the 3' LTR sequence carrying genes encoding Tax, HBZ, and other regulatory proteins. The CTCF binding site (vCTCF-BS) coincides with a sharp border in epigenetic modifications in the pX region of the provirus [15]. Therefore, it was hypothesized that CTCF bound in the pX region of the HTLV-1 provirus controls the epigenetic modifications and viral transcription. Furthermore, it was shown in T-cell lines and T-cell clones from patients that CTCF binding results in the HTLV-1 provirus forming abnormal chromatin contacts with sites in the host genome in *cis* as far as 1.4 Mb from the provirus [16]. However, a recent report suggested that epigenetic changes in the pX region occur independent of spontaneous HTLV-1 transcription and are CTCF-independent [17]. Therefore, it

remains unclear what role the binding of CTCF plays in viral replication and pathogenesis of HTLV-1 *in vitro* and *in vivo*.

In the current study, we established infected cell lines using HTLV-1 lacking the CTCF binding site and assessed the impact of inhibiting the interaction between CTCF and the viral genome through analysis of viral integration, epigenetic modifications, latency establishment, reactivation and transcription. We found that disruption of CTCF binding to the viral DNA caused a significant alteration of DNA methylation on both sense and antisense strands within the pX region of the provirus and in histone methylation both up- and downstream of the region. We also showed that by blocking CTCF binding in multiple HTLV-1 infected cell clones using shRNA expression, proviral gene expression was affected differentially and the overall effect was integration site dependent.

# Results

## Construction and characterization of HTLV-1ΔCTCF molecular clone

In order to study the role of the cellular CTCF protein in HTLV-1-mediated pathogenesis, two mutant proviral clones were created. HTLV-1ΔCTCF carries four point mutations within the consensus viral CTCF binding site (vCTCF-BS), which abolishes CTCF binding, but avoids introduction of mutations to the protein sequence of HBZ encoded from the opposite-strand of the viral DNA (Fig 1). We found that the mutations in vCTCF-BS eliminated CTCF binding as measured by electrophoretic mobility shift and chromatin immunoprecipitation (ChIP) assays (S1 Fig) [18].

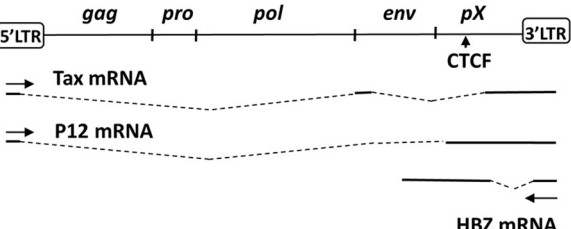

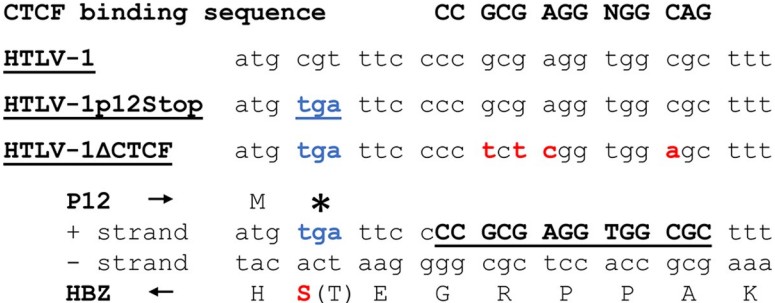

**Fig 1. HTLV-1 molecular clone with mutations in CTCF binding site (vCTCF-BS).** The top panel shows a schematic of the HTLV-1 proviral genome, and the positions of the pX domain (nucleotides 6619-8281), and the vCTCF-BS. Positions of sense-strand spliced *tax* and *p12* transcripts and the predominant antisense *hbz* transcript are depicted. The bottom panel shows the consensus CTCF binding sequence, and the adjacent nucleotide sequence in wild type HTLV-1. A control virus was constructed with a termination codon in p12 (blue) in place of alanine residue 77, designated HTLV-1p12Stop. A mutant virus was constructed with mutations (red) in the vCTCF-BS, as well as the p12 termination codon, designated HTLV-1ΔCTCF. The positions of mutations introduced in the sense reading frame of p12 and antisense reading frame of HBZ are shown at the bottom.

However, the changes in vCTCF-BS introduce mutations in p12, an HTLV-1 accessory gene encoded from the sense strand. To avoid producing a p12 gene product with multiple substitutions and potentially confounding results, a stop codon was introduced in p12, immediately upstream of the vCTCF-BS mutations, which deletes the last 23 amino acids of p12 and introduces a conservative Threonine to Serine change at codon 73 of HBZ (T73S)(Fig 1). Therefore, we also created a control clone with no change in vCTCF-BS, HTLV-1p12Stop, which contains only the p12 stop point mutation and Threonine to Serine change in HBZ. Previous data suggested that the C-terminus of p12 was not essential for its function, therefore we expected that HTLV-1p12Stop would behave the same as the wild type (WT) virus [19]. Moreover, several HTLV-1 isolates, and most simian T-cell leukemia virus type-1 isolates have a truncation of the C-terminus of p12 [20].

To assess if the point mutation (T73S) in HBZ results in a functional alteration, a plasmid encoding either wild type or mutant HBZ (T73S) was cotransfected into 293T cells with the Tax expression plasmid pS-Tax and the reporter pLTR-luc, or a plasmid expressing human p65 and the reporter pNF-KB-luc. HBZ (T73S) expression inhibited Tax- or p65-mediated transcriptional activation as effectively as wild type HBZ (S2 Fig), suggesting that the T73S mutation does not affect known functions of HBZ.

The importance of the C-terminal truncation in p12 was also assessed. Previous studies also showed that HTLV-1 p12 activates nuclear factor of activated T-cells (NFAT) mediated transcription [21]. To test whether the C-terminal truncation of p12 changes its function as an activator of NFAT-mediated transcription, we measured the luciferase expression driven by the NFAT promoter in Jurkat cells in the presence of p12, S3 Fig shows that the truncated p12 (p12stop) activates NFAT-mediated transcription as efficiently as wild type p12.

We next determined whether HTLV-1ΔCTCF or HTLV-1p12Stop mutant proviruses had altered viral gene expression. Transfection of either wild type HTLV-1 or mutant HTLV-1 proviral clones revealed no significant difference in the level and kinetics of Tax and HBZ mRNA expression (S4A and S4B Fig). Moreover, cells transfected with either HTLV-1ΔCTCF or HTLV-1p12Stop mutant viral clones produced similar levels of p19 Gag to that of wild-type HTLV-1 (S4C Fig). Taken together, these data indicate that the mutations at the vCTCF-BS and C-terminal truncation of p12 have no significant effect on the function of HBZ, p12, viral gene transcription, or production of virus particles *in vitro*.

## Deletion of the CTCF binding site of HTLV-1 results in expansion of DNA methylation in the pX region of the provirus

DNA methylation is an important mechanism for inactivating the HTLV-1 provirus, resulting in escape from the host immune response, and establishing the latent state [22, 23]. The proviral DNA sequences for the 5'LTR, *gag*, *pol*, *env* genes of HTLV-1 are heavily methylated, whereas the pX region and 3'-LTR DNA sequences are much less methylated [23]. Furthermore, this methylation pattern of the provirus does not change, whether or not there is expression of plus strand genes. The vCTCF-BS is located at the border of methylated and unmethylated proviral DNA sequences in HTLV-1 infected T-cell lines and freshly isolated PBMC from patients [15]. However, the role of methylation of the proviral DNA remains unknown, and it is also not clear how this pattern of proviral DNA methylation is regulated.

Because the vCTCF-BS coincides with the boundary of DNA methylation, it was reasoned that binding of CTCF introduces a barrier to block methylation extension to the downstream 3' region of the provirus [24]. To test this notion, we examined methylation of the DNA around the vCTCF-BS in a variant of Jurkat cells (JET cells), commonly used in HTLV-1 studies [25–27], and peripheral blood mononuclear cells (PBMCs) infected with HTLV-1ΔCTCF

or HTLV-1 or HTLV-1 p12Stop (S1 and S2 Tables). JET cells carry a tdTomato red fluorescent protein (RFP) under the control of a Tax responsive element. Multiple different IL2-dependent PBMC-derived cell lines had varying proportions of single or double positive or double negative CD4+ and CD8+ cells (S1 Table). However, no significant differences were seen in comparison of PBMCs infected with HTLV-1ΔCTCF or HTLV-1 or HTLV-1 p12Stop. Levels of *tax* and *hbz* RNA and production of p19 capsid protein per cell were slightly higher in PBMCs infected with HTLV ΔCTCF than those infected with wild type HTLV-1 or HTLV-1 p12Stop (S2 Table)

In order to analyze DNA methylation, we used next generation sequence analysis of bisulfite-treated proviral DNA from the bulk populations of JET cells and PBMCs. As expected, the viral genome upstream of the vCTCF-BS is heavily methylated to the same level in both wild type and the mutant proviruses, and a clear border is observed at the vCTCF-BS (Fig 2), consistent with a previous report [15]. Removal of CTCF binding dramatically enhanced methylation in the region downstream of the vCTCF-BS, including most of the pX region (Fig 2). This region of the provirus in HTLV-1ΔCTCF demonstrated up to 120 fold increase of methylation in more than 65% of CpG sites compared with WT HTLV-1 and HTLV-1p12Stop infected PBMCs (Table 1).

Similarly in JET cells (Table 1), HTLV-1ΔCTCF infected cells showed up to 31 fold increase of methylation in more than 55% of CpG sites compared with WT HTLV-1 and HTLV-1p12Stop proviral DNA. The greatest increase in methylation was observed in the region of 7023-7230 (P < 0.01) of the HTLV-1ΔCTCF provirus in both PBMCs and JET cells. However, the level of methylation in JET cells was less than that in PBMCs. It is unclear whether this is a

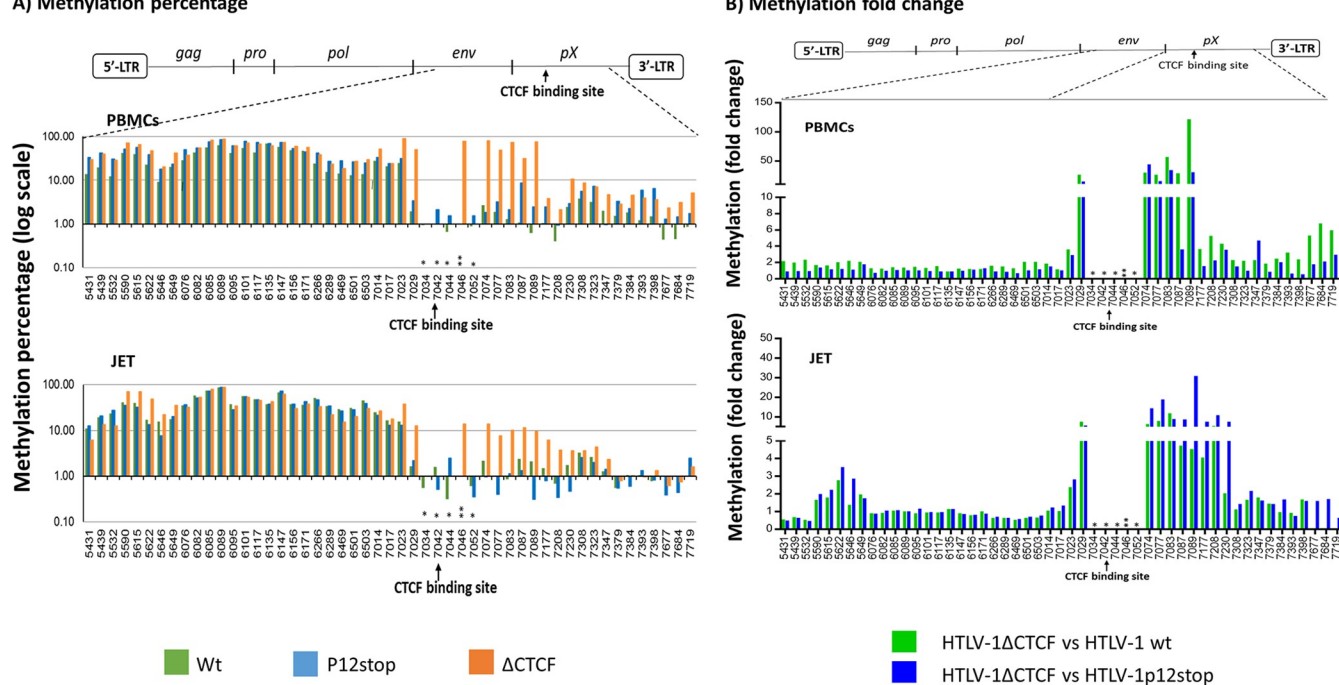

**Fig 2. Deletion of the CTCF binding site of HTLV-1 (vCTCF-BS) results in expansion of DNA methylation in the pX region of the provirus.** DNA methylation of the HTLV-1 provirus is presented as the percentage of methylated CpG (A) or fold change (B) (Y-axis) at the indicated locations of the viral DNA (X-axis). The schematic diagram of HTLV-1 provirus indicates the regions examined by bisulfite treatment and DNA sequencing as described in the Materials and Methods. Upper panel: HTLV-1 immortalized bulk population of PBMCs; lower panel: HTLV-1 infected bulk population of JET cells. CTCF binding site: 7041-7052 as indicated by an arrow. * Lost CpG sites, ** New CpG site due to the introduced mutations in vCTCF-BS. Nine CpG sites (5649-6076), 19 CpG sites (6503-7014), 8 CpG sites (7398-7615) are not shown because of unsuccessful PCR amplification in these regions after bisulfite treatment.

**Table 1. Summary of increased methylation in 5'LTR, *env*, *pX* region (sense) and 3'LTR/*hbz* (antisense) of HTLV-1ΔCTCF compared to wild type HTLV-1 or HTLV-1p12stop.**

| PBMC | | Fold increase (≥ 2 fold) | | Methylated CpG sites (≥ 2 fold increase) vs total examined CpG sites (%) | | Methylation level at each CpG site (%) (Average % is shown in each parenthesis) | | |
|---|---|---|---|---|---|---|---|---|
| HTLV-1 | | ΔCTCF / WT | ΔCTCF / p12stop | ΔCTCF / WT | ΔCTCF / p12stop | WT | p12stop | ΔCTCF |
| Sense Strand | 5'-LTR (166-679) | 2-15 | 2-15 | 15/40 (38%) | 16/40 (40%) | 0.2-1.5 (0.4%) | 0.2-5.2 (0.7%) | 0.1-5.3 (0.9%) |
| | *pX* (7023-7719) | 2-120 | 2-43 | 19/20 (95%) | 13/20 (65%) | 0.4-25 (2.8%) | 0.9-31 (4.7%) | 2-90 (25.5%) |
| Antisense Strand | 3'-LTR (8280-8959) | 2-6 | 2-7 | 14/48 (29%) | 15/48 (31%) | 0-0.35 (0.9%) | 0-11.7 (2.9%) | 0.1-9.5 (4.9%) |
| | *hbz* (6647-8235) | 2-10 | 2-6 | 35/50 (70%) | 20/50 (40%) | 0-66.3 (6.4%) | 0-53.9 (7.5%) | 0-85.5 (12%) |
| JET | | Fold increase (≥ 2 fold) | | Methylated CpG sites (≥ 2 fold increase) vs total examined CpG sites (%) | | Methylation level at each CpG site (%) (Average % is shown in each parenthesis) | | |
| HTLV-1 | | ΔCTCF / WT | ΔCTCF / p12stop | ΔCTCF / WT | ΔCTCF / p12stop | WT | p12stop | ΔCTCF |
| Sense Strand | 5'-LTR (166-679) | 20-60 | 20-140 | 2/40 (5%) | 3/40 (7.5%) | 0.2-7.6 (0.7%) | 0.1-2.2 (0.4%) | 0.2-13.2 (0.8%) |
| | *pX* (7023-7719) | 2-12 | 2-31 | 13/20 (65%) | 11/20 (55%) | 0-15.5 (2%) | 0.32-13.1 (1.7%) | 1-37 (6.6%) |
| Antisense Strand | 3'-LTR (8280-8959) | 2-40 | 2-30 | 12/48 (25%) | 10/48 (21%) | 0-2.5 (1.1%) | 0-3 (0.9%) | 0-5 (1.9%) |
| | *hbz* (6647-8235) | 2-5 | 2-5 | 11/50 (22%) | 20/50 (40%) | 0.2-20.9 (5.3%) | 0-20.3 (4%) | 0-14 (3.7%) |

result of cell type differences or differences in the duration of infection. The mutations introduced at the vCTCF-BS resulted in loss of four CpG sites (7034, 7042, 7044, 7052) and gain of one CpG site (7046); no methylation was seen at these sites except at the new CpG site (Figs 1 and 2).

We also examined methylation in the 5'LTR and 3'LTR. In the 5'LTR, the methylation was increased up to 15 fold in more than 38% of CpG sites of HTLV-1ΔCTCF infected PBMC proviral DNA as compared with the WT HTLV-1 and HTLV-1p12Stop provirus (S5A, S5B, and S5C Fig and Table 1). However, the methylation levels were less than 5% at those sites. There was no significant change of methylation in this region in HTLV-1ΔCTCF infected JET cells, compared to WT HTLV-1 and HTLV-1p12stop infected JET cells.

In the 3'LTR, we evaluated methylation on the anti-sense 3'LTR and pX region, which regulates the expression of HBZ [28]. Enhanced methylation (up to 10 fold) in HTLV-1ΔCTCF infected PBMC proviral DNA was found in more than 40% CpG sites of the pX region compared with WT HTLV-1 and HTLV-1p12Stop infected PBMCs (S5D, S5E, and S5F Fig and Table 1). In JET cells, HTLV-1ΔCTCF proviral DNA methylation exhibited an up to 40 fold increase in more than 21% CpG sites compared with WT HTLV-1 and HTLV-1p12Stop proviral DNA. Levels of antisense proviral DNA methylation were generally less than 10%.

In summary, our data indicate that the CTCF binding site of the HTLV-1 provirus is important to establish and maintain the boundary for DNA methylation on both sense and antisense strands of the pX region.

## Elimination of CTCF binding results in differentially increased histone methylation

Previous studies showed a sharp border in several important epigenetic modifications around the vCTCF-BS of the HTLV-1 provirus and hypothesized that the binding of CTCF controls neighboring epigenetic modifications and viral transcription [15]. Since our data indicated

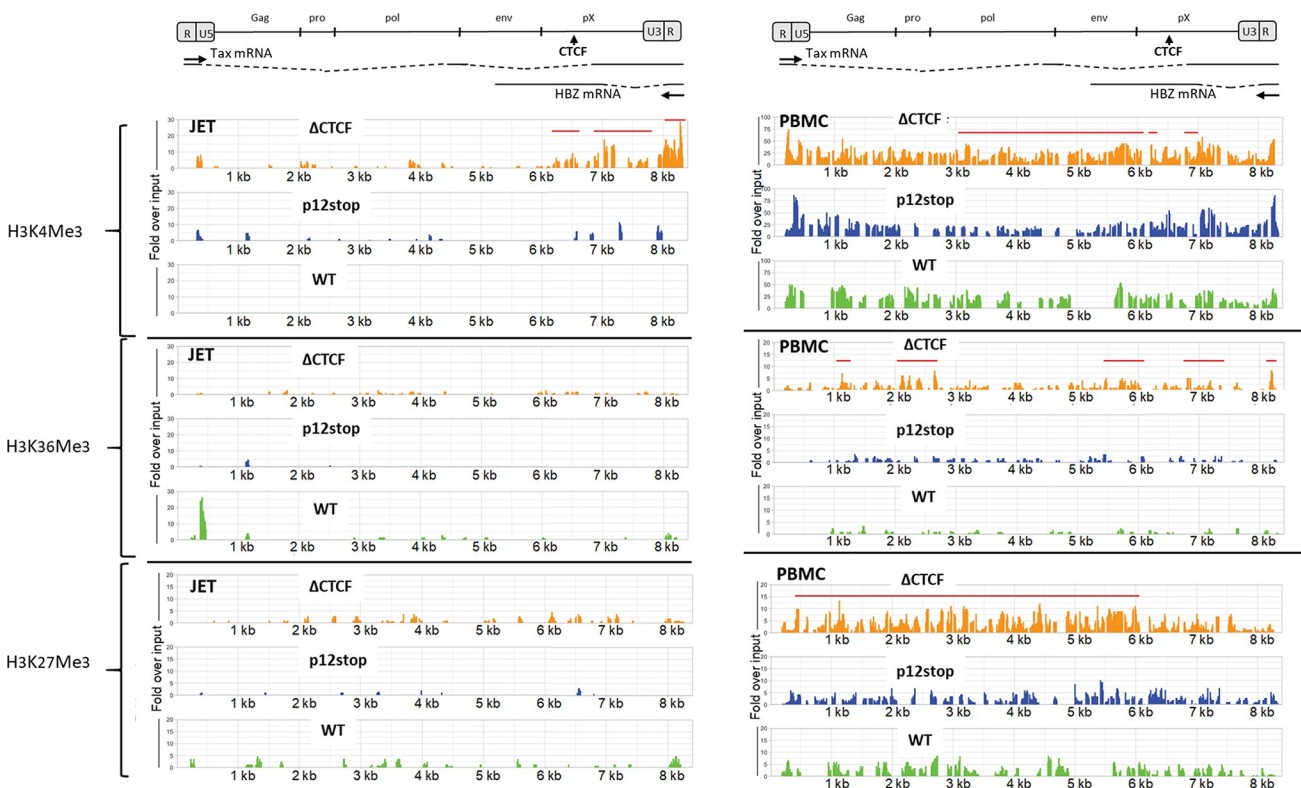

**Fig 3. Histone methylation is increased in HTLV-1ΔCTCF.** Plots show fold enrichment over input for the indicated chromatin modifications on HTLV-1 provirus DNA in the bulk populations of JET cells (left) or PBMCs (right). Cells infected with HTLV-1ΔCTCF (orange), HTLV-1-p12stop (blue), or WT HTLV-1 (green) virus were subjected to chromatin immunoprecipitation for H3K4Me3, H3K36Me3 or H3K27Me3. Called peaks in HTLV-1ΔCTCF that are significantly different from both HTLV-1p12Stop and WT HTLV-1 are indicated by horizontal red lines.

that CTCF binding was critical for maintaining the methylation boundary in the pX region and gene expression of the provirus, we questioned whether removal of the vCTCF-BS would alter epigenetic modifications of certain histone marks of the provirus. To address this question, we infected JET cells with either wild type HTLV-1 or the virus lacking the vCTCF-BS and collected cells 3 weeks post infection. In order to test if histone modifications are cell type dependent, we also infected PBMCs with the same viruses and examined immortalized primary T-cells. We first carried out a ChIP assay using CTCF Ab and performed qPCR targeting the vCTCF-BS of the provirus. The result of qPCR quantification confirmed that CTCF binding to the provirus of HTLV-1ΔCTCF is abrogated (S1A and S1C Fig). We also examined CTCF interaction with a human genome CTCF binding site [29]. As shown in S1B and S1D Fig, CTCF binds to this CTCF site in the human genome in all HTLV-1 infected cell lines.

We then assessed chromatin modifications by ChIP sequencing for methylation marks associated with active (H3K4Me3, H3K36Me3) or repressed (H3K27Me3) gene transcription. As shown in Fig 3, the level of H3K4Me3 was significantly increased both 5' and 3' of the CTCF site in HTLV-1ΔCTCF infected JET cells and PBMCs as compared to those infected with HTLV-1p12Stop or WT HTLV-1. In PBMCs, H3K4Me3 fold over input was approximately 2-4 times greater in HTLV-1ΔCTCF compared to HTLV-1p12Stop or WT HTLV-1, while in JET cells the H3K4Me3 level was 2-30 times greater than in control infection. Notably, the levels of H3K4Me3 were considerably higher over the entire provirus in PBMCs (25-75 fold over input) compared to JET cells (1-30 fold over input). Lower levels of histone

methylation marks, normalized to input, were found in JET cells compared to PBMCs, likely due to shorter duration of infection. The same data are shown in S6 Fig, without normalization for input values.

A more modest yet significant increase in H3K36Me3 was detected in some peak regions 5' and 3' of the CTCF site in PBMCs infected with HTLV-1ΔCTCF compared to HTLV-1p12Stop or WT HTLV-1. Finally, a significant increase in the level of H3K27Me3 was detected in nearly the entire proviral genome 5' to the CTCF site in PBMCs infected with HTLV-1ΔCTCF compared to HTLV-1p12Stop or WT HTLV-1, with levels 1-10 times fold higher over input values compared to the control infections.

In summary, these ChIPseq data demonstrate that loss of the vCTCF-BS in HTLV-1 results in increased histone methylation 5' and 3' of the CTCF site, suggesting that CTCF binding may act as a boundary element to epigenetic modification. This effect is similar to that seen with DNA methylation (Fig 2), except that the regions with altered histone methylation are much larger – up to 6 kb.

## Effect of CTCF binding on proviral integration site preferences

In order to assess whether the vCTCF-BS affected integration site preferences, we used sheared DNA from infected JET cells and PBMCs, which were subjected to linker ligation-mediated PCR and next-generation sequence analyses [5]. Unique integration sites (UIS) were identified as fragments with unique DNA fragment shear sites. Enumeration of reads at each UIS allowed determination of levels of clonality using the Gini index [30]. A consistently higher level of clonality was found in cells infected with HTLV-1p12Stop compared to HTLV-1ΔCTCF (Fig 4A).

In order to identify integration site preferences, we examined HTLV-1 integrations upstream or downstream of genes actively expressed in CD4+ lymphocytes and compared them to the prediction of completely random integration. Although there was a preference for integration within 100,000 nucleotides of active transcription start sites (ATSS), no differences were seen with JET cells or PBMCs infected with WT HTLV-1, HTLV-1p12Stop, or HTLV-1ΔCTCF (Fig 4B). A preference for integration within 100,000 nucleotides of CTCF sites (Fig 4C) and CpG islands (Fig 4D) was also observed with WT HTLV-1, HTLV-1p12Stop and HTLV-1ΔCTCF infected cells, but no significant differences were detected with those cells. We further examined the integration site preferences based on clone abundance for integration sites within 1000 nucleotides (S7A, S7B, and S7C Fig) or 10,000 nucleotides (S7D, S7E, and S7F Fig) of ATSS, CpG islands and cellular CTCF binding sites, respectively. Once again, no significant differences were identified in comparisons of WT HTLV-1, HTLV-1p12Stop, or HTLV-1ΔCTCF JET cells or PBMCs.

## Inhibition of CTCF binding to the HTLV-1 provirus affects proviral gene transcription

In light of the results on DNA and histone methylation, we next asked if disruption of CTCF binding would also affect gene expression of the provirus. Since gene expression of HTLV-1 proviruses is highly dependent on the site of integration, we examined the effect of CTCF in clonal cell lines carrying a single provirus [31]. For this purpose, we infected JET cells with WT HTLV-1, HTLV-1p12Stop, or HTLV-1ΔCTCF, and established virus infected cell lines. Each cell line carried one copy of the provirus at a unique insertion site, and exhibited no baseline gene expression (Table 2). We then knocked-down expression of CTCF by stably expressing shRNA targeting CTCF, which decreased the level of CTCF by a mean of 70% compared to control shRNA in these cell clones (Fig 5A-a). By decreasing the amount of CTCF available

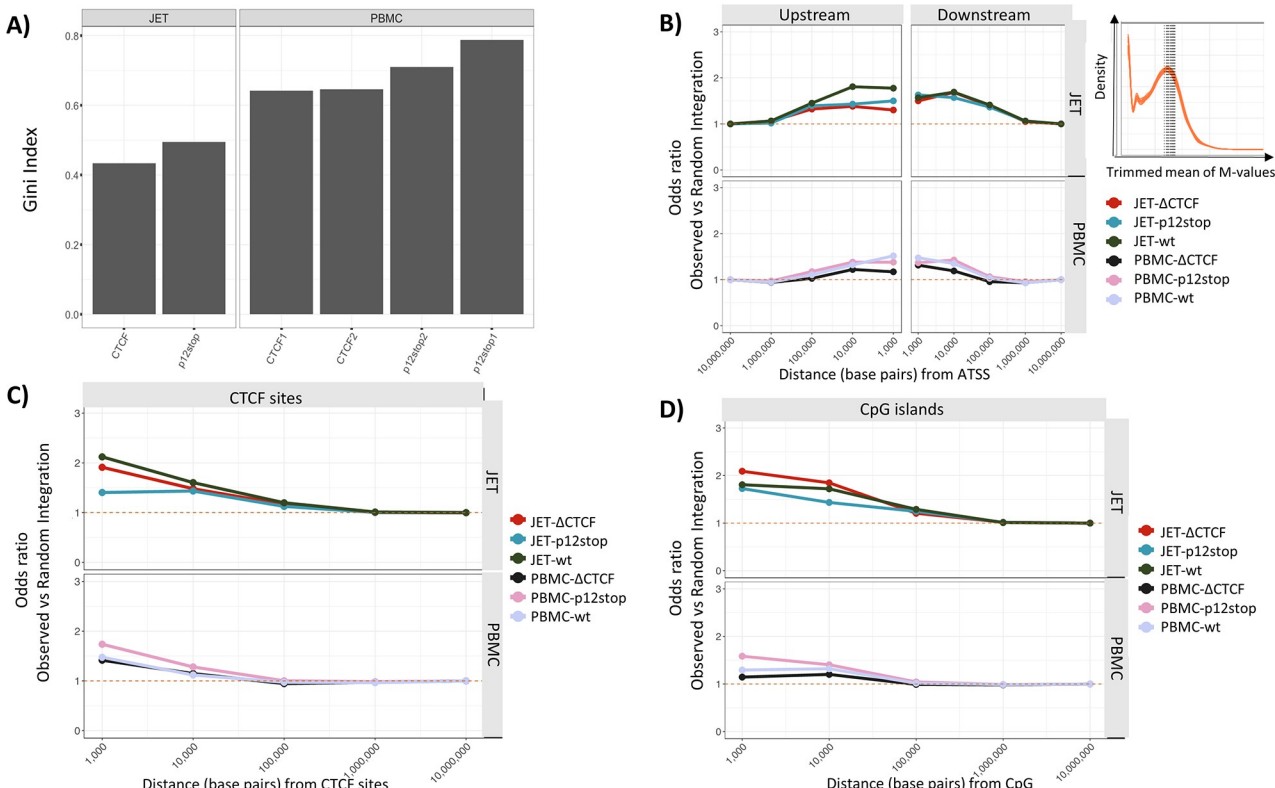

**Fig 4. Effect of CTCF-BS Mutation on HTLV-1 Integration Sites.** A) Clonality of HTLV-1 and HTLV-1ΔCTCF integration sites in JET cells and PBMCs, as measured by the Gini index. Inconsistent results were obtained with HTLV-1 WT infected cells. B-D). Frequency distribution of observed WT HTLV-1, HTLV-1p12stop, and HTLV-1ΔCTCF integration sites compared to random expectation within indicated window sizes (base pairs) in JET cells and PBMCs. B) Frequency of integration positions relative to upstream or downstream active transcription start sites (ATSS). The inset shows the transcriptional frequency of active genes in normal CD4+ T lymphocytes, with the shaded gene showing that for Fox P3. C) Frequency of HTLV integration sites relative to cellular CTCF sites. D) Frequency of HTLV integration sites relative to CpG islands.

for binding to the provirus in the cells, we were able to assess the effect of CTCF binding on the pattern of reactivation of proviral transcription. We next treated the cells with phorbol myristyl acetate (PMA) plus Ionomycin and determined activation of viral gene expression by measuring Tax mediated red fluorescence protein (RFP) production using the IncuCyte live cell image system. Of 15 WT HTLV-1 clonal cell lines, 10 (66%) exhibited significantly higher levels of RFP and *tax* mRNA when CTCF was knocked down as compared with control shRNA. Representative clonal lines with higher RFP levels (Type I, #4 and #9) and unchanged RFP (Type II, #11 and #15) are shown in Fig 5B (left panels) and the remainder in S8 Fig. We confirmed the levels of sense strand transcription by measuring *tax* mRNA by quantitative real-time RT-PCR (Fig 5B middle panels), which were consistent with RFP levels in all clonal lines. There were no significant differences in levels of *hbz* mRNA expression in the presence of CTCF versus control shRNAs (Fig 5B, right panels). Levels of *hbz* mRNA are typically $10^3 - 10^4$ lower than that of *tax* and *gag-pol* mRNA [32–34]. To test if the viral gene expression can be reactivated through a more natural pathway, we treated cells with CD3/CD28 antibodies. As expected, similar results were observed except that CD3/CD28 stimulation was much weaker (S9 Fig).

We also examined methylation of the viral DNA around the vCTCF-BS in these clonal lines using next generation sequence analysis of bisulfite-treated proviral DNA. The CTCF knockdown induced a significant increase of methylation in most type I but not type II cell lines (Fig 6). Four type I lines (67%) exhibited a higher average fold increase in CpG methylation than

Table 2.  **HTLV-1 infected Jet cell clones.**  Positions of HTLV-1 integration sites are shown for each cell clone.

| Type I | Integration site | |
|---|---|---|
| Clones | Chromosome | Position |
| 1 | 14 | 65024208 |
| 2 | 16 | 17059283 |
| 3 | 5 | 48895079 |
| 4 | 13 | 60212040 |
| 5 | 2 | 104381109 |
| 6 | 3 | 185768470 |
| 7 | 1 | 74594663 |
| 8 | 7 | 57772359 |
| 9 | 11 | 65937735 |
| 10 | 18 | *27355914* |
| Type II | Integration site | |
| Clones | Chromosome | Position |
| 11 | 5 | 25127602 |
| 12 | 4 | 150572223 |
| 13 | X | 79356997 |
| 14 | 22 | 39239471 |
| 15 | 10 | 55546159 |

seen in type II lines. Therefore, elevated Tax expression in the majority of type I cell lines positively correlates with enhanced methylation in the pX region when CTCF is down regulated.

Infectious virus production was also examined from the same four WT HTLV-1 clonal JET cell lines treated with control or CTCF shRNA knock-down (Fig 5A-b). Briefly, clonal cells were co-cultivated with Jurkat cells carrying the HTLV-1 LTR fused to a firefly luciferase gene and luciferase activity was measured 48hr after PMA and ionomycin treatment. The clonal cell lines with higher levels of Tax and RFP also produced higher amounts of infectious virus, as measured by luciferase activity, in the presence of CTCF shRNA compared to control shRNA. No differences were seen in the clonal cell lines with unchanged Tax and RFP. Importantly, CTCF knockdown had no effect on cell viability (S10 Fig).

To test whether the effect of CTCF on proviral gene expression was mediated through interactions outside the provirus, we stably expressed CTCF or control shRNA in clonal JET cell lines containing one copy of latent HTLV-1ΔCTCF provirus (Fig 5A-c and 5C). As shown in Figs 5C and S11, CTCF and control shRNA expressing cells showed very similar profiles of *tax* mRNA and RFP production. Taken together, these results suggest that loss of CTCF binding at the vCTCF-BS leads to increased viral gene expression, and that this effect is not due to loss of CTCF binding elsewhere in the host genome. These results further support a role for CTCF in modulating proviral gene expression and suggest that the effect is dependent on the viral integration site, which is consistent with previous observations that the pattern of gene expression of HTLV-1 proviruses is determined by the flanking genomic sequences of the host [31, 35]. Interestingly, we did not observe any cell clone in which CTCF knock down significantly reduced proviral gene expression, implying that CTCF mainly functions as an insulator to suppress transcription of the provirus.

## Latency establishment and reactivation of HTLV-1 does not depend on CTCF binding

CTCF binding regulates latency establishment and reactivation of lytic DNA virus replication [36]. Since our studies showed that CTCF suppressed gene expression of the HTLV-1 provirus

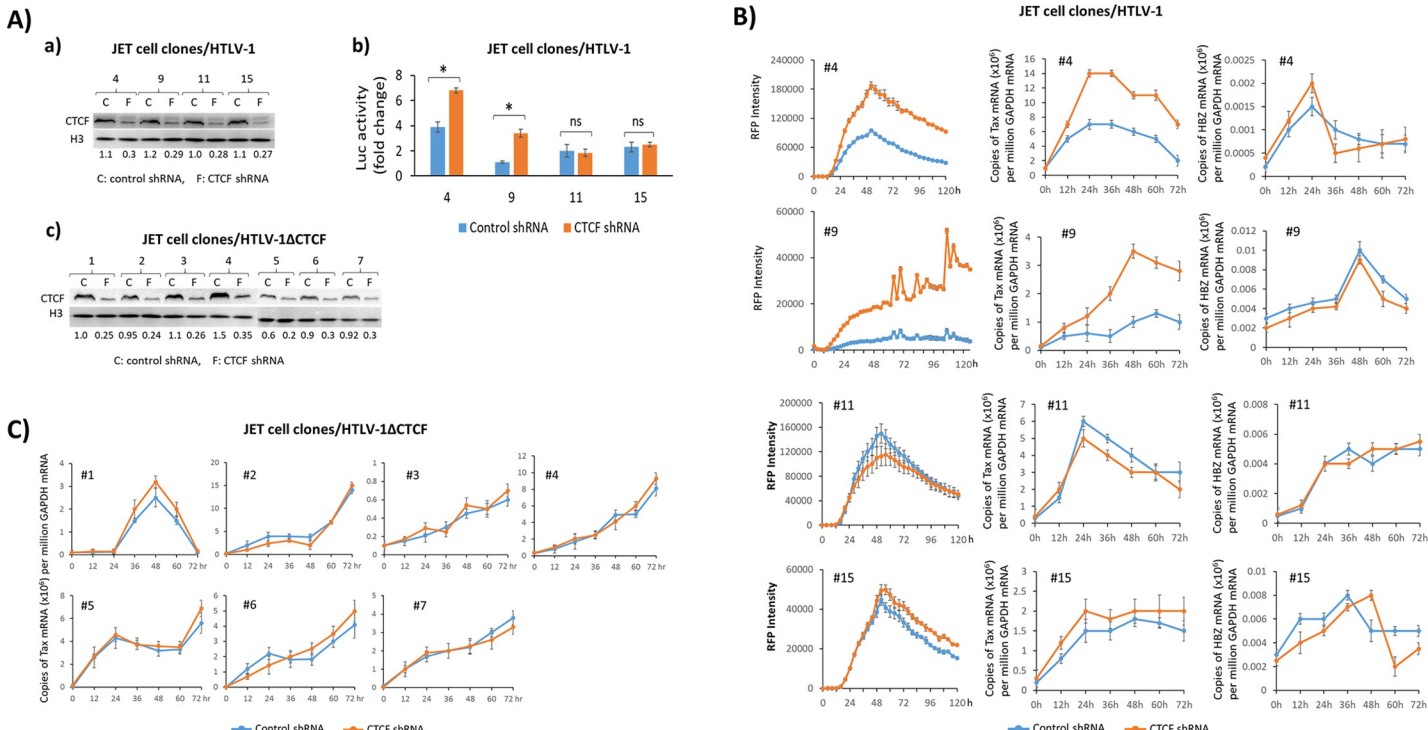

**Fig 5. Inhibition of CTCF binding to HTLV-1 provirus affects proviral gene transcription.** Clonal HTLV-1 (A-a and B) or HTLV-1ΔCTCF (A-c and C) infected JET cell lines were generated, each carrying a single, latent provirus at a unique integration site. The infected JET cells from panels B, and C were collected and cellular CTCF was detected by Western blotting using anti-CTCF antibody. Histone H3 was also blotted as a loading control. The amount of CTCF was measured by densitometry quantification using BioRad image lab software, normalized to the value of histone H3 and is indicated under each lane. A-b). The JET cells from Fig 5B were cocultured with Jurkat cells expressing firefly luciferase driven by the HTLV-1 LTR (LTR-Luc) in the presence or absence of PMA/Ionomycin for 48 hr and viral infectivity was determined by measuring luciferase activity and presented as a fold change compared with the values of untreated cells (t test: $^*p < 0.05$; ns: not significant). B & C). Viral gene expression was activated with PMA/Ionomycin and monitored by measuring Tax mediated RFP production using the IncuCyte live cell image system. The levels of *tax* and *hbz* mRNA were also measured by qRT-PCR. Total copy number of each mRNA was determined using plasmid DNA standards and normalized to $10^6$ copies of GAPDH mRNA. Cell clone number of HTLV-1 infected and HTLV-1ΔCTCF cells are indicated in each panel. Each experiment was repeated three times and a representative result with three replicates is shown with standard errors.

in the majority of our cell clones, we asked if the CTCF binding plays a part in establishing latency and reactivation. To address this question, we infected JET cells with WT HTLV-1, HTLV-1p12Stop, and HTLV-1ΔCTCF, respectively, and selected single cell clones via limiting dilution. The infected cell clones were then examined for the expression of RFP, the indicator of Tax expression. RFP positive cell clones were counted as lines that carry actively transcribed viral genome, and the RFP negative cells as lines that carry latent viruses. Long range PCR (5'LTR to 3'LTR) was performed to examine the RFP negative cells to determine which lines carry a full-length viral genome, and those with deletions were excluded. Approximately 300 cell clones from each infection were obtained in three independent experiments. The percentage of latent infected RFP negative HTLV-1ΔCTCF infected cells was not significantly different from WT HTLV-1 (P = 0.22) or HTLV-1p12Stop (P = 0.12) infected cells (Fig 7A).

We also determined whether disruption of the CTCF binding would affect reactivation of the latent provirus. RFP negative cell lines were activated with 48 hrs of treatment with PMA/Ionomycin. RFP positive and negative cells were then enumerated (Fig 7B). The proportion of RFP positive cells (reactivated cells) was not significantly different between cell lines infected with HTLV-1ΔCTCF compared to those infected with WT HTLV-1 (P = 0.47) or HTLV-1p12Stop (p=0.14). This result shows that abolition of CTCF interaction with HTLV-1 DNA

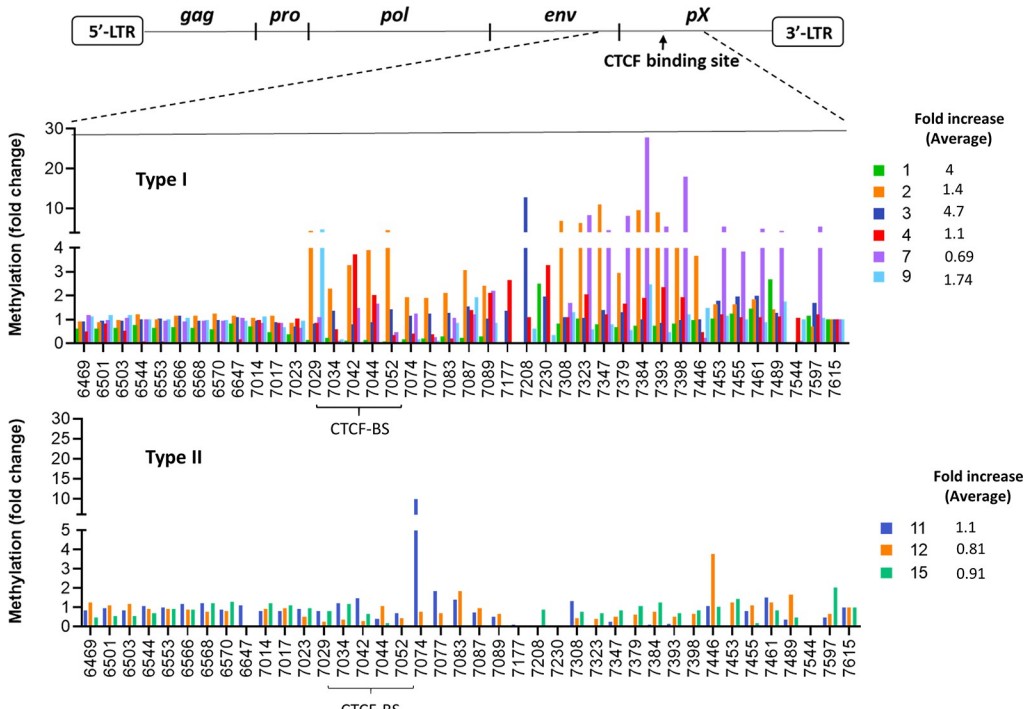

**Fig 6. CTCF knock down by shRNA in clonal JET cell lines results in expansion of DNA methylation in the pX region of the provirus.** DNA methylation of the HTLV-1 provirus is presented as fold change, CTCF vs control shRNA (Y-axis) at the indicated locations of the viral DNA (X-axis). Average fold increase is calculated from nt 7029 to 9615. The schematic diagram of the HTLV-1 provirus indicates the regions examined by bisulfite treatment and DNA sequencing as described in the Materials and Methods. Upper panel: Type I clonal cell lines. Lower panel: Type II clonal cell lines.

does not change viral latency establishment and reactivation and indicates that these processes are probably independent of CTCF-mediated regulation of gene expression.

## Discussion

Gene expression of the HTLV-1 provirus is strongly impacted by the surrounding genomic sequences [31]. The discovery that the HTLV-1 provirus binds CTCF led to the hypothesis that CTCF binding regulates gene expression of the provirus [15]. A recent study revealed that binding of CTCF to the provirus altered host gene expression by forming chromatin loops with the neighboring host genome and caused deregulation of host transcription at a large number of loci in each infected host [16]. However, the role of CTCF in regulation of gene expression of the HTLV-1 provirus remains poorly understood. Our previous study showed that CTCF did not affect HTLV-1 LTR transactivation, viral particle production, or immortalization capacity *in vitro* [18].

In the current study, we showed that elimination of the vCTCF-BS not only causes remarkable enhancement of methylation in the pX region, shifting the border of this epigenetic modification closer to the 3'-LTR, but also significantly increases the methylation on the antisense strand encoding HBZ. It was postulated that by forming a barrier to epigenetic modifications, CTCF may cause or perpetuate differential transcription of the 5′ and 3′ parts of the provirus, resulting in persistent expression of HBZ, which is required for clonal persistence [13]. Although how methylation of the proviral DNA is regulated is still unknown, and the precise mechanism of how methylation controls gene expression of the provirus is also not well

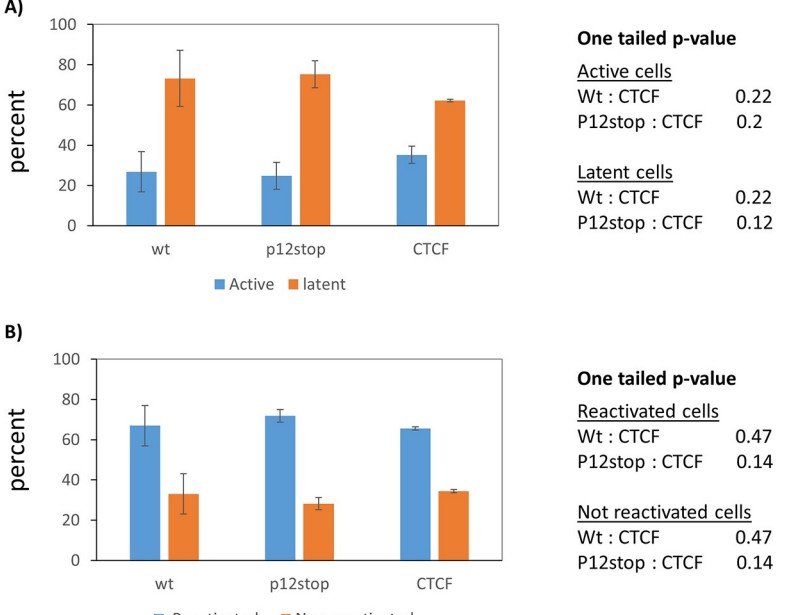

**Fig 7. Latency establishment and reactivation of HTLV-1 does not depend on CTCF binding.** JET cells were infected by coculturing with lethally γ-irradiated 729B/HTLV-1, 729B/HTLV-1p12stop or 729B/HTLV-1 ΔCTCF and single infected cell clones were selected via limiting dilution. RFP (the indicator of Tax expression) positive or negative cells were examined and percentages of latent (RFP negative) and active (RFP positive) HTLV-1 infected cells are shown in panel A. RFP negative cells were activated with PMA/Ionomycin for 48hr. RFP positive and negative cells were counted and the result from three experiments is presented with standard errors in panel B.

understood, it is clearly established that the 5'-LTR plays a key part in controlling sense strand transcription [31]. Based on the observation that the *gag*, *pol* and *env* sequences of the provirus are heavily methylated, Tanigichi et al speculated that DNA methylation first occurs in the *gag*, *pol* and *env* regions and then extends in the 5' and 3' directions *in vivo*; when the 5'-LTR becomes methylated, viral transcription is silenced [37]. Miura et al. later showed that the plus strand expression was silenced even though the 5'LTR was hardly methylated, implying that DNA methylation at the 5'LTR is not the sole factor that suppresses the viral transcription [17]. Intriguingly they further demonstrated that the pattern of hypermethylation in internal sequences of proviruses in various cell lines did not change regardless of the sense strand expression. This finding indicated that the methylation pattern of the internal DNA is an innate characteristic of the provirus and independent of integration location. However, we cannot exclude the possibility that expanded methylation may exist in some infected cells, but not be detected due to the limitations of the methods utilized. It is possible that suppression of sense strand transcription requires methylation of the 5'-LTR in conjunction with hypermethylation of the internal sequences of the provirus. Thus, extending methylation beyond the established border in the pX region could potentially influence the regulatory function of the 5'-LTR for sense strand transcription. However, the final effect of the extended hypermethylation in the pX region likely depends on a combination of methylation status in both the internal and LTR regions. Interestingly increased methylation in the 5' and 3'-LTR was also detected in the HTLV-1ΔCTCF provirus but at lower levels, which seems to agree with the speculation of Taniguchi and colleagues that methylation starts in the internal sequences then extends to the 5' and 3' LTR [37].

In other contexts, CTCF loss mediates unique DNA hypermethylation surrounding CTCF binding sites in a variety of human cancers [38]. In the prostate, the Igf2-H19 locus

experiences DNA methylation with CTCF downregulation at multiple intergenic CTCF sites [39]. A study of seven immortalized cell lines and twelve normal cell cell types showed that DNA methylation was linked to CTCF binding at 40% of sites across the genome [24]. It was suggested that variation in CTCF occupancy may derive from a complex regulation of specific interaction partners [40]. Others suggested that loss of CTCF expression might activate DNA methyl transferases [41]. However, we found no correlation of changes in HTLV-1 gene expression in response to demethylating drugs with the phenotypes identified in our type I and type II infected JET cell lines (Fig 5 and not shown). Alternatively, DNA methylation may prevent CTCF binding [42, 43]. The correlation between methylation site results and positive strand gene expression (Figs 5 and 6) suggests a significant relationship between these findings. However, additional studies are required to identify the precise mechanisms involved.

Whole-genome studies have shown that exons are more highly methylated than introns and transitions in the degree of methylation occur at exon-intron boundaries, possibly suggesting a role for methylation in regulating splicing [44]. Shukla et al demonstrated that CTCF binding results in pausing of RNA pol II and changes in the kinetics of Pol II movement, thus influencing splicing [45]. They further showed that whether exons are favorably included or not in mature transcripts depends on the position of CTCF binding sites. Interestingly Satou et al recently reported that CTCF down regulation by shRNA leads to reduced p30 expression and postulated that CTCF plays a role in the regulation of HTLV-1 RNA splicing because the vCTCF-BS is near the splice junction of the p30 transcript [15]. In the current study, we observed a significant increase of proviral methylation in the pX region, and Tax expression in two thirds of our clonal cell lines when CTCF is knocked down. Intriguingly the vCTCF-BS is located immediately upstream of the third exon of the *tax/rex* transcript, thus RNA splicing could be a step where CTCF exerts its impact. However, the exact mechanism of how CTCF binding influences processing of viral transcripts remains to be elucidated. In addition, Satou's report also revealed that some sequences near the vCTCF-BS exert CTCF-dependent enhancer blocking activity, thus loss of the vCTCF-BS could relieve the block which could consequently promote transcription driven by the 5' or 3'LTR if an enhancer is involved in expression of either proviral strand.

Although it remains unclear if the methylation expansion in the pX region on the sense strand could directly affect *hbz* transcription, the increased methylation in the HBZ coding region on the antisense strand could reduce HBZ expression. Therefore, we predicted that the enhanced methylation in the HBZ coding sequence caused by ablation of the vCTCF-BS could result in less expression of HBZ, which in turn could lead to more transcription of the sense strand from the 5'-LTR. It is well known that HBZ protein is expressed at a low level in naturally infected cells and the low level HBZ is thought to be necessary for the persistence of HTLV-1 *in vivo* [46]. Recently, Billman et al showed that *hbz* is not expressed in all cells at all times, even within one clone of cells, and most cells contain only 0~5 transcripts. Moreover, *hbz* mRNA mostly remains in the nucleus with a half-life of 4.4 hours and is exported from the nucleus at a very low rate, resulting in even lower levels of HBZ protein [47]. However, in the current study, we did not detect any significant reduction *of hbz* mRNA in our cell clones which showed increased expression of Tax protein when CTCF was knocked down. One explanation could be that the *hbz* mRNA level may not accurately reflect the level of HBZ protein, which inhibits Tax expression by competing for binding of the transcription co-factor CREB.

Within the context of gene regulation, DNA methylation does not function in isolation. Instead, DNA methylation and histone modifications have a combinatorial effect on gene expression. There is emerging evidence that DNA methylation can affect the methylation states on accompanying histones in chromatin, while the histone lysine methylation state of

chromatin can in turn influence modification of the DNA itself [35]. Consistently, we observed that the removal of the vCTCF-BS not only increased nearby viral DNA methylation, but it also led to increased methylation of nearby histone H3, and also histones separated by up to 6 kb. These findings indicate that CTCF binding to the provirus plays an important role in regulating histone modification as well as viral DNA methylation.

Interestingly both active (H3K4Me3, H3K36Me3) and repressive (H3K27Me3) histone marks were enhanced in HTLV-1ΔCTCF infected cells. Although H3K27Me3 is known to be a characteristic of silencers, unlike H3K9Me3 which remains silenced all the time and prevents multiple transcription factors (TFs) from binding, H3K27Me3 still allows certain genes to be activated through TF binding in a different cell state [48]. Furthermore, this function is cell type specific and highly context dependent. It can act as a silencer in one cell line but a super enhancer in another cell line [49–55]. Thus it is not a determinant factor of silencers on its own [56]. It has been shown that H3K27Me3 and H3K4Me3 frequently coexist in genes expressed at low levels in embryonic stem cells (ES), and it was proposed that bivalent domains silence developmental genes in ES cells while keeping them poised for activation [48]. It is likely that viral gene expression is also regulated simultaneously by both active and repressive histone marks. The opposite effect of these two functions could enable a fine-tuning of the viral gene expression and provide great sensitivity and robustness to the regulatory circuitry of plus and minus strand expression, especially for *tax* and *hbz* transcripts.

Surprisingly we only observed a clear border for H3K4Me3 modification in HTLV-1 infected JET cells but not the other histone methylation marks. Significant increases of both H3K4Me3 and H3K36Me3 due to the vCTCF-BS deletion were also detected on both sides of the vCTCF-BS. Therefore the role of CTCF as a boundary element for histone methylation may not be as important as for DNA methylation. Both DNA methylation and histone modification have been reported to affect gene expression of the HTLV-1 provirus [17, 23, 37]. Future studies should investigate how these two mechanisms interplay to collectively regulate proviral gene expression.

Changes in HTLV integration site specificity did not explain the alterations in proviral DNA methylation. No significant differences were found in comparison of the proximities of integration sites for WT HTLV-1, HTLV-1p12Stop, and HTLV-1ΔCTCF relative to start sites of transcriptionally active genes, cellular CTCF binding sites, and CpG islands. However, we did see a higher clonality index in HTLV-1p12Stop compared to HTLV-1ΔCTCF infected JET cells and PBMCs. This may reflect a global difference in integration site specificity.

Satou et al initially analyzed the effect of vCTCF-BS deletion by CRISPR-Cas9 in one ATL clonal cell line (one viral copy per cell) and observed a slight change in the border of H3K36Me3 and a significant reduction of *hbz* mRNA after culturing cells with IL-2 [15]. But Miura et al. later reported that removal of the vCTCF-BS did not affect the epigenetic modifications and transcriptional activity of the provirus in two HTLV-1 patient-derived T-cell clones, each carrying one copy of the provirus lacking the vCTCF-BS [17]. Their methylation analysis was based on ten or fewer clones in each experiment. In order to more completely evaluate the role of CTCF binding, we established multiple HTLV-1 infected cell clones, each carrying a single copy of the provirus integrated at a unique chromosomal location. In addition, we used next generation sequence methods to analyze several thousand reads for each potential CpG site. By examining these infected cell lines, we were able to investigate the effect of CTCF binding on reactivation of the provirus in various genomic environments of the host. Notably, in the majority of clonal cell lines, decreased levels of CTCF led to increased methylation in the pX region and *tax* gene expression, demonstrating that the CTCF binding plays a critical part in the regulation of viral DNA methylation and gene expression in the majority of cell clones infected with WT HTLV-1. However, because the actions of CTCF at a given

location depend on the local context involved in site-specific interactions between CTCF, transcriptional cofactors, and the primary DNA sequence (for example, enhancers and silencers), removal of CTCF does not invariably lead to changes in chromatin looping and epigenetic marks flanking the CTCF binding site [57, 58]. Therefore, differences in proviral integration sites may explain why one-third of our clonal lines and two of Miura's clonal lines were unaffected by CTCF knockdown, and also why ablation of the vCTCF-BS did not result in methylation increase in all CpG sites of the pX region in the infected PBMCs and JET cells, each of which contain about 3000 unique integration sites.

Some limitations of our current work are the use of a single virus with a mutation in the vCTCF-BS, studies restricted to Jurkat cells and primary human lymphocytes, lack of dynamic continuous monitoring of *hbz* expression in JET cell lines, and use of a surrogate for continuous monitoring of *tax* expression based on RFP expression.

Nevertheless, the important role of CTCF in modulating viral gene expression is underscored by the fact that viral gene expression was not affected by CTCF knockdown in HTLV-1ΔCTCF cell lines, but was amplified in the majority of WT HTLV-1 cell lines, strongly suggesting that CTCF binding imposes a suppressive effect on viral gene expression. This is consistent with the role of CTCF as an insulator either blocking transcriptional enhancement or forming an epigenetic barrier [59]. The enhanced gene expression of the provirus caused by knockdown of CTCF in the cell clones could be due to increased levels of the activating histone mark, H3K4Me3, as seen with the loss of the vCTCF-BS in HTLV-1ΔCTCF. These findings highlight a novel mechanism in which a retrovirus hijacks a host protein to regulate its gene expression.

## Materials and methods

### Cell lines and culture

JET WT35 (referred as JET cells in this paper) is a subline of Jurkat cell expressing tdTomato red fluorescent protein (RFP) under the control of 5 times tandem repeat of Tax responsive element (TRE) [27]. HTLV-1 infectivity indicator cell line: JURL cells carry HTLV-1 LTR U3 driven- firefly luciferase, and RSV driven-renilla luciferase [60]. The 729B cells are human lymphoblastoid cells used extensively for HTLV-1 expression [61].

### Plasmids and cloning

The infectious HTLV-1 ACH molecular clone was used to generate all HTLV-1 infected cell lines for this study [62]. The HTLV-1 ACH molecular clone contains the Neo^R gene and has been previously described [63]. Site-directed mutagenesis of HTLV-1 was used to generate HTLV-1ΔCTCF and HTLV-1p12Stop molecular clones. HTLV-1ΔCTCF contains four point mutations within the consensus CTCF binding site (vCTCF-BS) while avoiding introduction of mutations to the opposite-strand coding sequence of the *hbz* gene (Fig 1). However, the vCTCF-BS mutations result in changes of the p12 coding sequence, a sense transcribed HTLV-1 accessory gene [19]. In lieu of producing a p12 gene product with multiple substitutions and potentially confounding results, an additional mutation was introduced in the p12 gene, immediately upstream of the vCTCF-BS mutations that results in deletion of the carboxy terminal 23 amino acids of p12 (Fig 1). HTLV-1p12Stop contains only the p12 stop codon point mutation, and thus serves as a control for potential effects of the p12 deletion. Plasmids were verified by sequence analysis.

### Establishment of HTLV-1 producer cell lines

Stable 729 HTLV-1 producer cell clones were generated by nucleofection of 729B cells with 2 µg of WT HTLV-1, HTLV-1ΔCTCF, or HTLV-1p12Stop plasmid using an Amaxa Cell Line

Nucleofector Kit V in accordance with the manufacturer suggested protocols (program X-001; Amaxa, Cologne, Germany). Nucleofected cells were then subjected to G418 selection (1 mg/ mL; Life Technologies, Carlsbad, CA). An HTLV-1 ELISA was used to confirm p19 Gag production in G418 selected cell lines. Cell lines with p19 Gag production were then single cell selected via limiting dilution. HTLV-1ΔCTCF and HTLV-1 p12Stop mutations were confirmed via Sanger sequencing. The p19 Gag ELISAs were performed on single cell clones, and those with comparable p19 Gag production were selected for future study. The 729B/HTLV-1, 729B/HTLV-1p12Stop or 729B/HTLV-1ΔCTCF are three virus producer cell lines used in this study.

## Generation of immortalized PBMCs

Human PBMCs were isolated from freshly collected whole blood using the Ficoll-Paque PLUS (GE Healthcare Bio-Sciences AB, Uppsala, Sweden) density gradient medium. Two million PBMCs were cocultured with one million lethally γ-irradiated (100Gy) 729B/HTLV-1, 729B/ HTLV-1p12, or 729B/HTLV-1/ΔCTCF cells in RPMI 1640 medium supplemented with 20% FBS, 10 U/mL of recombinant human interleukin-2 (Roche Diagnostics GmbH, Mannheim, Germany), glutamine, and antimicrobials in a 24-well plate. After three weeks live cells were counted and p19 in the medium was measured by ELISA every week to monitor infection and immortalization of PBMCs by HTLV-1. Immortalized PBMCs were cultured for 4-5 months and stored at -80˚C. DNA was isolated using DNeasy blood tissue kit (Qiagen) for methylation, epigenetic modification and integration site studies.

## Bisulfite treatment and DNA sequencing

DNA was extracted from infected cell lines and PBMCs with DNeasy Blood and Tissue kit (Qiagen, Cat #51106). Two μg of purified DNA was subjected to bisulfite treatment using Epitect Bisulfite kit (Qiagen, Cat #59104) with slight modifications of the manufacturer's protocol. Bisulfite primers (BSP) were designed using meth primer software (https://www.urogene.org/ cgi-bin/methprimer/methprimer.cgi) or manually to cover ~2 kb close to CTCF-BS in HTLV-1 genome (Fig 2). PCR was performed using Epitaq HS (Takara, Cat #R110B) with following conditions 95˚C for 3min ; 95˚C for 10 s, annealing temperature for 20 s, 72˚C for 20 sec (40 cycles); 72˚C for 5 min. The primer sequence and annealing temperatures are shown in S4 Table. Amplified PCR products were gel purified using Gel DNA extraction kit (Zymo Research, Cat #D4007) and further processed for adapter ligation and DNA deep sequencing as described by McDonald et.al [64]. CpGs covered by 30 or more reads were considered for further analysis. Methylation percentage was calculated based on number of methylated reads vs number of unmethylated reads.

For the LTR region, two step PCR was performed in order to separate 5' LTR and 3'LTR region. To amplify 5' LTR, a BSP forward primer was designed in the start of the 5'U3 LTR region and a BSP reverse primer was designed in the start of *gag* region. For the antisense 3'LTR region forward primer was located in the HBZ coding region whereas the reverse primer was in the U5 portion of the 3' LTR region. PCR was performed as described above and the products were gel purified and 1 ul was used as template for next PCR using the primers listed in S4 Table. The same PCR conditions for 15 cycles were used for these steps. After the PCR products were gel purified, adapters were ligated, and samples sequenced as describe above.

Amplicon bisulfite sequencing data were analyzed similarly to McDonald et al. [64, 65]. Adaptors and poor quality sequences (quality less than 20) were trimmed using TrimGalore (v0.5.0). Trimmed sequences were mapped to the HTLV-1 DNA sequence (NC_001436.1)

with or without the mutations at the CTCF-BS using Bismark [65] using the non-directional flag. Methylation calls for the appropriate strand were then extracted from resultant bam files and converted to bedGraph files using bismark_methylation_extractor with the paired-end and bedGraph flags [65]. Only CpGs covered by at least 30 reads were considered.

## ChIP assay and ChIP-seq

Infected JET cells or immortalized human PBMCs were cross-linked with 1% formaldehyde at room temperature for 10 min. Glycine was then added to a final concentration of 0.125 M to quench the reaction. After wash with PBS, cells were aliquoted and frozen at -80˚C. Five million cells were lysed in 300 ul SDS lysis buffer (1% SDS, 10mM EDTA, 50 mM Tri-HCl pH8.0) plus protease inhibitors (Thermo Scientific, Cat #A32953) for 20 minutes at 4˚C and DNA was sheared to an average size 200-300 bp using Bioruptor. The fragmented chromatin samples were collected after removal of cellular debris by centrifugation at 13000 rpm at 4°C for 10 min and frozen for subsequent immunoprecipitation. Two micrograms of each antibody (Cell Signaling, #4909: H3K36Me3 Ab; #9751:H3K4Me3 Ab; #9733:H3K27Me3 Ab) were added to 300 ul sheared chromatin diluted with 700 ul ChIP dilution buffer and the mixture was incubated overnight at 4˚C with rotation. Then protein A beads (Diagenode, Cat #K0614006) were added and incubation continued for overnight at 4˚C with rotation. The beads were washed with the following buffers: a) Low salt wash buffer (0.1% SDS, 1% Triton X-100, 2 mM EDTA, 20 mM Tris-HCl pH8.0,150 mM NaCl). b) High salt wash buffer (0.1% SDS, 1% Triton X-100, 2 mM EDTA, 20 mM Tris-HCl pH8.0, 500 mM NaCl). c) LiCl wash buffer (0.25 M LiCl, 1% NP40, 1% deoxycholic acid, 1 mM EDTA, 20 mM Tris-HCl pH8.0). d) 20 mM Tris-HCl, pH 8.0 (TE). e) TE. DNA was eluted with freshly prepared elution buffer (1% SDS, 0.1 M NaHCO$_3$) and crosslinks were then reversed at 65°C in the presence of 0.2 M NaCl and proteinase K overnight. Finally, DNA was isolated with QIAquick DNA purification kit (Qiagen, Cat #28104). The specificity of each chromatin immunoprecipitation was confirmed by examining a positive and negative control gene for each ChIP by qPCR. The control primers were purchased from Cell Signaling and each set of control genes were as follows: RPL30 exon 3 (positive #7014) and MyoD1 exon 1 (negative #4490) for H3K4Me3; MyoD1 (positive) and RPL30 (negative) for H3K27Me3; GAPDH intron2 (positive #4478) and GAPDH promoter (negative #4471) for H3K36Me3.

For ChIP sequencing analysis, immunoprecipitated DNA was blunt ended, added with "A" base to 3' end and ligated with sequencing adapters to each side. The fragments were size selected to 200-600 base pairs with Ampure beads (Beckman-Coulter, AMpure XP A63880) and PCR amplification for 15 cycles with primers incorporating unique dual index tag for multiplexing. The resulting libraries were sequenced using the Illumina HiSeq3000 as single reads extending 50 bases. Each DNA sequence was mapped using NovoAlign (V3.04.06) algorithm to the human reference genome GRCh38.d1.vd1 containing the HTLV-1 sequence (https://gdc.cancer.gov/about-data/data-harmonization-and-generation/gdc-reference-files). Peak detection was performed using the model-based analysis of the ChIP-seq data (MACS version 2.1.1.20160309). Histone modification peaks were defined by significant enrichment of each signal over input-DNA peaks at a P cutoff value of $10^{-5}$. For visualization, HTLV-1 genome regions were extracted from ChIP-seq data (bam files) using samtools (version 1.1.1). HTLV-1 mapped reads were normalized to total mapped reads for each ChIP sample, sequence fragments were extended to 200 bp, fold over matched input calculated, and graphing performed using R (version 4.0.2 GUI 1.72, RStudio 1.3.1073, packages tidyr, dplyr, sqldf, ggplot2, stats, and grid.extra). ChIP-seq data is available at GEO (acquisition number GSE172194).

### Integration site mapping and data analysis

**Identification of integration sites for infected clonal JET cells.** Inverse long PCR (IL-PCR) was performed to amplify the genomic DNA flanking the 3′LTR according to a modified method described previously [66]. In brief, the genomic DNA was digested by *Pst*I, self-ligated by T4 ligase, and then digested by *Mlu*I. Long PCR amplification of the linearized DNA was performed using the PrimeSTAR GXL DNA polymerase (TaKaRa, Cat #R050A) according to the manufacturer's protocol. Primer sets for IL-PCR analysis are listed in S3 Table. IL-PCR products were isolated from agarose gels, purified, and subjected to nested PCR. Amplified nested PCR fragments were subcloned into pUC19 digested with Sma-I (NEB) and sequenced to obtain provirus integration sites downstream of the 3′ long terminal repeat.

**Identification of integration sites for infected JET cells and PBMCs.** Mapping Integration sites of HTLV-1 infected JET cells and PBMCs was conducted by following Firouzi's protocol [67]. Briefly a total of 10 μg of DNA was sheared into 300-700 bp by sonication with a Covaris S2 Sonicator. DNA ends were end-repaired using T4 DNA polymerase, DNA polymerase I Klenow fragment, and T4 polynucleotide kinase (New England Biolabs, Cat #M0201S). Addition of an adenosine at the 3′ ends of the DNA was performed using Klenow fragment 3′ to 5′ exo- (New England Biolabs, Cat #M0212S). A partially double-stranded DNA adaptor was ligated to the DNA ends using T4 DNA ligase (New England Biolabs, Cat #M0202S). Each adaptor contains a tag of 8-bp random nucleotides which enables DNA fragments to be uniquely marked. Nested PCR was performed to amplify the ligated PCR products by using F1 and R1 for the first PCR and F2 and P5 for the second PCR. Finally, the flow cell binding sequence P7 with a specific 6-bp sample index was added to the second PCR products. DNA was cleaned using a Qiaquick PCR purification kit.

The nested PCR library was sequenced on an Illumina NovaSeq Instrument as paired 2x150 bp reads. Read pairs were filtered such that they must contain the expected viral sequence (GAAATTTAGTACACA) and could not include a series of artifactual sequences using a python script. This script allowed for a one base mismatch to account for sequencing error. Sample demixing was then performed using fgbio (1.0.0) (https://github.com/fulcrumgenomics/fgbio). The two independently sequenced libraries for each sample were merged using samtools (1.10) [68] and aligned to a custom reference genome sequence containing HGRCh38 and HTLV1 (D13784.1) using bwa-mem (v0.7.17; with parameters -t 4 -Y). Duplicates were then marked using Picard (2.22.9). Reads were then split out into two groups based on integration direction using samtools (flag filter parameters -f147 -F3584 OR -f131 -F3600). In order to determine viral integration locations, mosdepth (0.2.5; -Q 20) [69] was used to generate a per base position coverage report of reads meeting the filtered criteria across the genome. An R script was then used to identify deviations from a coverage of 0 indicating the start of a viral integration site. To account for situations where a homozygous germline deletion might artificially split an integration site into two entries, integration sites within 150 bp were merged.

**Creating random integration sites.** A list of approximately 1 million randomly inserted integration sites was obtained using bedtools random (2.29.2; -l 10) [70] for GRCh38. In order to ensure that a randomly inserted integration site could be observed in a realistic dataset we assessed regions of the genome where sequencing coverage was possible to observe and limited random sites to this subset of the genome as follows. We calculated the coverage of 18 normal WGS samples using mosdepth (0.2.5; -Q 20). Integration sites were retained if they were within regions of the genome where $> = ⅔$ of our normal samples achieved a coverage of 20x.

**Calculating odds ratio/relative risk.** The genomic distances between features and integration sites were calculated in order to determine odds ratio and relative risk. Briefly we

investigated three features in relation to integration sites: CD4+ active transcription start sites, CTCF binding sites, and CpG islands. CD4+ active transcription start sites were defined as genes with trimmed mean of m-values (TMM) greater than 10 in CD4+ cells from the GSE60424 dataset. CTCF sites were derived from CD4+ cells originating from Cuddapah et al [71]. CpG sites were derived from the "strict" set from the CpG island DB https://bioinfo2.ugr.es/CpGislandDB/datasets/. In all cases feature coordinates were transformed to GRCh38 using liftover where applicable. Genomic distances between these features and integration sites were determined using bedtools closest (2.29.9; -d). Odds ratios and relative risk were calculated between experimental integration sites and random integration sites using R for different genomic distances. In all calculations the random integration sites were sampled to match the number of integration sites in the experimental set.

**Calculating Gini index.** The Gini index [30] was calculated using the R package Desctools (https://andrisignorell.github.io/DescTools/authors.html).

## Generation of HTLV-1 infected JET cell lines and CTCF knock down by shRNA

JET cells were cocultured with lethally γ-irradiated 729B/HTLV-1, 729B/HTLV-1p12Stop or 729B/HTLV-1ΔCTCF for 3 days and sorted for red fluorescence protein (RFP) positive cells. After another 21 days of culture, portions of sorted cells were saved for the studies of methylation, epigenetic modifications, and integration sites. The remaining cells were further single cell selected via limiting dilution, and cell clones that did not express RFP, but could be activated with PMA/Ionomycin to produce RFP, were selected as latently infected cells. Digital droplet PCR and long-range PCR (5'LTR to 3'LTR) were performed to screen for cell clones that contain only one copy of a full-length viral genome. To knock down cellular CTCF, cell clones that carried one copy of provirus and were latent in viral gene expression were transduced with the lentiviral vector (pLKO.1-puro) encoding CTCF (GCAAGGCAAGAAAT GCCGTTA) or control shRNA (GTTCCGTCATAGCGATAACGA). Raltegravir (1uM, Sell-eckchem, Cat #MK-0518) was added to prevent new infection [72]. After puromycin selection, surviving cells were examined for CTCF knock down by Western blot, and the viral copy number in each cell line was quantified by ddPCR. Integration sites were mapped by Inverse long PCR (IL-PCR) and Sanger DNA sequencing. Cell clones that were latent in viral gene expression and contained one copy of provirus integrated at a unique site were chosen for further study.

## PCR, RT-qPCR and ddPCR

To verify the mutations in infected cell lines, DNA was isolated from 729B/HTLV-1, 729B/HTLV-1p12, and 729/HTLV-1ΔCTCF producer cell clones, as well as the immortalized PBMCs, and infected JET cells using the DNeasy Kit (Qiagen, Valencia, CA). Standard PCR followed by Sanger sequencing for vCTCF-BS mutation verification was performed for each newly generated producer cell clone and cocultured immortalized hPBMCs for 16 wks. The vCTCF-BS primer sets (S3 Table) and the following PCR conditions were utilized for PCR amplification: 95˚C for 3 min followed by 35 cycles of 95˚C for 15 sec, 60˚C for 10 sec and 72˚C for 1 min. Amplified PCR products for each sample were then gel purified using the Qiagen gel extraction kit (Qiagen, Valencia, CA) and submitted for Sanger sequencing.

Digital droplet PCR (ddPCR) was performed with DNA from infected cells, and the viral copy number was quantified using the primers and probes as indicated in S3 Table [73].

RT-qPCR was performed with RNA extracted with the RNeasy kit (Qiagen) and the spliced forms of *tax* or *hbz* mRNAs were measured using the one-step RT-PCR kit (BioRad, Hercules,

CA). The plasmids encoding the cDNAs for HBZ or Tax were utilized as standards to determine the amount of each mRNA and the results were normalized to GAPDH mRNA [74].

## Live cell imaging and analysis

JET cells containing HTLV-1 proviruses and expressing CTCF or control shRNA were seeded at a density of 50,000 cells/well in a 96 well plate pre-coated with poly lysine (Millipore-Sigma, St. Louis, MO, USA). RPMI media, lacking phenol red, supplemented with 10% FBS, 100 U pencillin/streptomycin and 1ug/ml puromycin was used. To count live cells, Cytolight Rapid reagent (Essen Bioscience, Cat #4705) which stains live cells and produces green fluorescence, was added based on the manufacturer's protocol. Prior to imaging, PMA (50 ng/ml) and Ionomycin (1 µM) were added to activate the cells. Then cells were continuously imaged for RFP (*tax* expression) and green fluorescence (live cells) every 3 hours for 96 hrs using an IncuCyte live cell S3 analysis system and IncuCyte S3 v2017A software (Essen Bioscience) with default parameters set for 96 well TPP plates using a 10 X objective. The IncuCyte software was used to calculate the mean integrated intensity for RFP of all live cells from five non overlapping images. The experiments were performed in triplicate.

## Statistics

P values were determined by 1-tailed t-tests.

## Supplemental methods

Additional plasmids for assessing effects of mutations in HBZ and p12.
   Transfection, quantification of luciferase activity and Western blotting.

## Supporting information

**S1 Fig. The mutations in vCTCF-BS eliminate CTCF binding to the HTLV-1 but not cellular DNA.** Chromatin immunoprecipitation (ChIP) of HTLV-1 infected Jurkat cells and PBMCs was performed using CTCF antibody and immunoprecipitated DNA was quantified by qPCR. A, C) ChIP with CTCF antibody and qPCR targeting the HTLV-1 vCTCF-BS or *gag* gene. B, D) ChIP with CTCF antibody or IgG and PCR targeting the cellular CTCF binding site (see S3 Table for primer sequences).
(TIF)

**S2 Fig. The mutation (T73S) in HBZ does not affect its function of inhibiting activity of transcription activator Tax and P65.** 293T cells were transfected with the reporter plasmid pLTR-Luc or pNF-KB-luc (200 ng, firefly luciferase), pS-Tax or pHA-65 and pCNF-HBZ (wild type or T73S), and the internal control plasmid (50 ng) pTK-Renila (Renila luciferase). Cells were collected and lysed at 48 hr after transfection. Firefly luciferase activity was measured and normalized to the value of Renila luciferase. Expression of each protein in the transfected cells was examined by Western blotting.
(TIF)

**S3 Fig. The truncation mutation in *p12* of HTLV-1 proviral clone does not affect the function of p12.** Jurkat cells were transfected with pNFAT-luc and plasmid expressing wild type, truncated P12 (p12stop) or the vector pCNF. Cells were activated with PMA (50 nM) at 12 hr post transfection and collected after 30 hr activation. pTK-Renila plasmid was cotransfected with each set of indicated plasmids and firefly luciferase activities were normalized to the value of Renila luciferase. Activation of NFAT promoter is presented as the ratio of luciferase activity

of PMA treated to non treated cells. Expression of p12 protein was examined by Western blotting.
(TIF)

**S4 Fig. Mutations at CTCF binding site of HTLV-1 proviral clone does not affect the viral transcription and production.** A & B). 293T cells were transfected with equal amounts of HTLV-1 plasmid DNA (2 ug DNA) containing either wild type or mutations as indicated and collected every 12 hr. After extraction of total RNA from the transfected cells, *Tax* or *hbz* mRNA was measured by qRT-PCR and normalized to *GAPDH* mRNA. C). Viral production was examined by measuring p19 in the medium at 48hr after transfection using ELISA.
(TIF)

**S5 Fig. DNA methylation change in 5'LTR and 3'LTR of the HTLV-1ΔCTCF compared to wild type HTLV-1 and HTLV-1p12stop provirus.** A, D). DNA methylation is presented as the percentage of methylated CpG (Y- axis) in 5'LTR (A), 3'LTR and *hbz* on anti-sense strand (D) of HTLV-1ΔCTCF compared to wild type HTLV-1 or HTLV-1p12stop provirus at the indicated locations of the viral DNA (X- axis). Upper panel: HTLV-1 immortalized PBMCs; lower panel: HTLV-1 infected JET cells. B, C). DNA methylation is shown as a fold increase of methylated CpG (Y- axis) in 5'LTR of HTLV-1ΔCTCF HTLV-1 compared to wild type HTLV-1 or HTLV-1p12stop provirus in PBMCs (B) and JET cells (C). E, F). DNA methylation is shown as a fold increase of methylated CpG in 3'LTR and *hbz* on anti-sense strand of HTLV-1ΔCTCF compared to wild type HTLV-1 or HTLV-1p12stop provirus in PBMCs (E) and JET cells (F). Upper panel: 3'LTR; lower panel: *hbz*. CTCF binding site: 7041-7052 as indicated by an arrow. * Lost CpG sites, ** New CpG site due to the introduced mutations in vCTCF-BS.
(PDF)

**S6 Fig. Epigenetic modifications in the wild type and ΔCTCF HTLV-1 provirus.** Histone methylation marks of HTLV-1 provirus were analyzed by ChIP-seq in the bulk populations of JET cells (left) or PBMCs (right). LTR reads were randomly mapped to the 5′ LTR or 3′ LTR, because their sequences are identical. Five million JET cells and ten million PBMCs were utilized for ChIP-seq.
(TIF)

**S7 Fig. Effect of CTCF-BS mutation on HTLV-1 integration frequency.** Frequency of integration according to a clonal abundance within 1 kb (A-C) and 10 kb (D-F) of active transcription start sites (ATSS), CpG islands and cellular CTCF binding sites.
(TIF)

**S8 Fig. Inhibition of CTCF binding to HTLV-1 provirus affects proviral gene transcription.** Clonal HTLV-1 infected JET cell lines were generated, each carrying a single, latent provirus at a unique integration site. Viral gene expression was activated with PMA/Ionomycin and monitored by measuring Tax mediated RFP production (Y- axis) using the IncuCyte live cell image system.
(PDF)

**S9 Fig. Inhibition of CTCF binding to HTLV-1 provirus affects proviral gene transcription.** Clonal HTLV-1 infected JET cell lines were generated, each carrying a single, latent provirus at a unique integration site. Viral gene expression was activated with CD3/CD8 antibodies and monitored by measuring Tax mediated RFP production (Y-axis) using the IncuCyte live cell image system.
(PDF)

**S10 Fig. CTCF knockdown by shRNA had no effect on cell viability.** Clonal HTLV-1 infected JET cell lines were treated with PMA/Ionomycin to activate viral gene expression in the presence of Cytolight Rapid reagent which stains live cells and produces green fluorescence. Viral gene expression and number of live cells were measured by monitoring Tax mediated RFP and green fluorescence product, respectively using the IncuCyte live cell image system.
(TIF)

**S11 Fig. Inhibition of CTCF binding to HTLV-1ΔCTCF provirus does not affect proviral gene transcription.** Clonal HTLV-1ΔCTCF infected JET cell lines were generated, each carrying a single, latent provirus at a unique integration site. Viral gene expression was activated with PMA/Ionomycin and monitored by measuring Tax mediated RFP production using the IncuCyte live cell image system.
(TIF)

**S12 Fig. CTCF knock down by shRNA in JET clonal cell lines results in expansion of DNA methylation in the pX region of the provirus.** DNA methylation of the HTLV-1 provirus is presented as the percentage of methylated CpG (Y- axis) at the indicated locations of the viral DNA in pX region (X-axis). The schematic diagram of HTLV-1 provirus indicates the regions examined by bisulfite treatment and DNA sequencing as described in the Materials and Methods. The number of each clonal cell line is labeled on the right side of the figure.
(PDF)

**S1 Table. CD4 and CD8 expression by Immortalized PBMCs with HTLV-1.**
(TIF)

**S2 Table. Expression of *tax* and *hbz* mRNA, and capsid protein p19 from Immortalized PBMCs with HTLV-1.**
(TIF)

**S3 Table. Primers used in ddPCR, qPCR, qRT-PCR, IL-PCR and DNA deep sequencing.**
(TIF)

**S4 Table. Primers used in Bisulfite sequencing.**
(TIF)

## Author Contributions

**Conceptualization:** Lee Ratner.

**Data curation:** Xiaogang Cheng, Ancy Joseph, Victor Castro, Alice Chen-Liaw, Zachary Skidmore, Malachi Griffith, Jacqueline E. Payton, John R. Edwards, Lee Ratner.

**Formal analysis:** Xiaogang Cheng, Ancy Joseph, Victor Castro, Alice Chen-Liaw, Zachary Skidmore, Daniel A. Rauch, Malachi Griffith, Jacqueline E. Payton, John R. Edwards, Lee Ratner.

**Funding acquisition:** Lee Ratner.

**Investigation:** Xiaogang Cheng, Ancy Joseph, Victor Castro, Alice Chen-Liaw, Zachary Skidmore, Michael P. Martinez, Patrick Green, Malachi Griffith, Jacqueline E. Payton, John R. Edwards, Lee Ratner.

**Methodology:** Xiaogang Cheng, Ancy Joseph, Victor Castro, Alice Chen-Liaw, Zachary Skidmore, Daniel A. Rauch, Grant A. Challen, Malachi Griffith, Jacqueline E. Payton, John R. Edwards, Lee Ratner.

**Project administration:** Lee Ratner.

**Resources:** Takaharu Ueno, Jun-ichi Fujisawa, Michael P. Martinez, Patrick Green.

**Supervision:** Lee Ratner.

**Validation:** Lee Ratner.

**Writing – original draft:** Xiaogang Cheng, Lee Ratner.

**Writing – review & editing:** Xiaogang Cheng, Ancy Joseph, Victor Castro, Alice Chen-Liaw, Zachary Skidmore, Takaharu Ueno, Jun-ichi Fujisawa, Daniel A. Rauch, Grant A. Challen, Michael P. Martinez, Patrick Green, Malachi Griffith, Jacqueline E. Payton, John R. Edwards, Lee Ratner.

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
