## [Decision Letter · Decision Letter 0]

23 Jan 2021

Dear Dr. Ratner,

Thank you very much for submitting your manuscript "Epigenomic Regulation of Human T-Cell Leukemia Virus by Chromatin-Insulator CTCF" for consideration at PLOS Pathogens. As with all papers reviewed by the journal, your manuscript was reviewed by members of the editorial board and by several independent reviewers. The reviewers appreciated the attention to an important topic. Based on the reviews, we are likely to accept this manuscript for publication, providing that you modify the manuscript according to the review recommendations. In particular, we would ask you to carefully address the comments from reviewer 1.

Sincerely,

Bryan R. Cullen

Associate Editor

PLOS Pathogens

Richard Koup

Section Editor

PLOS Pathogens

Kasturi Haldar

Editor-in-Chief

PLOS Pathogens

orcid.org/0000-0001-5065-158X

Michael Malim

Editor-in-Chief

PLOS Pathogens

orcid.org/0000-0002-7699-2064

Reviewer Comments (if any, and for reference):

Reviewer's Responses to Questions

**Part I - Summary**

Reviewer #1: In this manuscript, Cheng et al. investigate potential functions of the CTCF binding site in the HTLV-1 genome. The authors generated a CTCF mutant proviral clone that disrupts CTCF binding, which was then used to infect PBMCs and JET cells. Since the CTCF mutations also result in p12 mutations, the authors introduced a stop codon in p12 upstream of the CTCF mutations. A proviral clone with only the p12 deletion was used as a control. The authors found increased DNA methylation in the genomes of delta CTCF viruses in both PBMCs and JET cells, mainly downstream of the CTCF binding site. There was also an increase in both activating (H3K4me3) and repressive (H3K27me3) histone marks in delta CTCF viruses in PBMCs and JET cells. Finally, mutation of the CTCF binding site did not impact proviral integration site preferences, but did influence proviral gene expression. Knockdown of CTCF with shRNA enhanced viral gene expression in JET cells infected with wild type HTLV-1, but not delta CTCF viruses.

Overall, the results suggest that the CTCF binding site serves as an epigenetic barrier to limit the spread of methylation to the 3’ LTR and thus protect HBZ from inactivation, however it is difficult to understand why there were no noted changes in HBZ expression associated with loss of the CTCF binding site. Furthermore, the results from this study are directly opposed to those from a recent study (Miura et al. 2018 Wellcome Open Research 3:105), and thus additional experiments are warranted to rigorously substantiate the conclusions. The presented data convincingly implicate CTCF in the suppression of plus strand HTLV-1 gene expression; however, it remains unclear if the underlying mechanism is due to epigenetic changes (i.e. DNA and histone methylation) in the HTLV-1 genome caused by loss of the CTCF binding site.

Reviewer #2: Recent work demonstrated the existence of a CTCF binding site in the HTLV-1 provirus (Satou, PNAS). This is a nice paper reporting the effects on DNA methylation and expression of HTLV-1 proviral DNAs, upon mutating the CTCF binding site. The usual switch from highly methylated upstream (in the “sense” direction), to poorly methylated downstream, is mostly lost in the mutant. This is a very intriguing result (predicted by early studies and here confirmed to be true). Histone marks were also affected.

All the work here is done with cell lines infected in vitro, so we are looking at integration sites established by recent infection. The work was done in pooled JET cells and immortalized clones of PBMC cells.

The authors next looked effects of KD of CTCF on transcription, using infected JET clones that are initially silent, and then activated with PMA and ionomycin, plus/minus KD of CTCF. The KD gave increased viral gene expression in a subset of the clones (not all).

There was no effect of the mutation on the frequency of establishment of latently infected clones or the reactivation from latency. Similar results to this were recently reported by some of the authors (and are cited here: Martinez, Retrovirology). There were no changes in integration site preferences.

Somewhat conflicting results on the effects of CTCF BS mutation on viral chromatin were recently reported (Wellcome Open Research. 2018;3:105) in a different setting.

This is a generally nice story on an interesting site in the HTLV-1 genome. There’s a lot of work presented, and the findings are generally convincing and properly interpreted. There are limitations to the settings (cells, timing) that are not ideal but unavoidable, and just should be admitted firmly somewhere in discussion.

Reviewer #3: In this manuscript, Cheng and Joseph et al. generated and examined an HTLV-1 mutant virus that eliminates the binding site for the epigenomic insulator CCCTC-binding protein (CTCF) located in the pX region of the viral genome. Using a variety of state-of-the-art techniques and clonally infected T-cells, the authors clearly show that mutation of the CTCF-binding site (CTCF-BS) does not disrupt viral gene expression. Moreover, establishment of latency and reactivation from latency do not seem to depend on the CTCF-BS. However, DNA deep sequencing of bisulfite treated DNA revealed that the viral genome is heavily methylated upstream of the CTCF-BS, while it is only weakly methylated downstream of the CTCF-BS. Interestingly, disruption of the CTCF-BS leads to a dramatically enhanced methylation in the region downstream of the CTCF-BS. ChIP sequencing data underline the insulating role of the viral CTCF-BS since disruption of CTCF-BS results in differentially increased histone methylation of the HTLV-1 provirus. Using next generation sequencing, the authors show that disruption of the CTCF-B does not impact proviral integration site preferences of HTLV-1. However, analysis of a series of clonally infected cells by qPCR and live cell imaging revealed that inhibition of CTCF binding affects viral gene expression in ca. 2/3 of all investigated clones, leading to increased viral gene expression. Finally, the authors show that cells infected with delta-CTCF-BS virus do not differ in establishment of viral latency nor do they differ in viral reactivation from latency, suggesting that control of latency may be independent of CTCF-binding.

This is a very interesting manuscript which clearly elucidates the relevance of the CTCF-BS in HTLV-1 proviral DNA using a broad panel of up-to-date methods. The manuscript is well-written and the experiments are clearly described. The authors comprehensively analyzed the newly established proviral clones and the respective controls. The experimental results strongly support the conclusions of the authors. Due to the importance of the data showing that mutation of the CTCF-BS does not impact latency establishment and reactivation (Fig S10), these data should be shown in the main part of the manuscript.

**Part II – Major Issues: Key Experiments Required for Acceptance**

Reviewer #1: 1) The authors need to characterize the HTLV-1 immortalized PBMCs used in the study. Are these cells CD4+, IL-2 dependent, etc.? What is the baseline expression of Tax, HBZ and Gag mRNA and protein in PBMCs immortalized with WT, p12Stop and delta CTCF viruses? Based on the authors’ model, differences in viral gene expression should be apparent in the absence of a functional CTCF binding site in the viral genome.

2) It is unclear if the methylation studies presented in Figs. 2 and 3 represent single cell clones or the bulk population of PBMCs and JET cells. This needs to be clarified by the authors. If these experiments were performed with the bulk population of cells, they should be repeated with single cell clones with defined viral integration sites. Why is the overall histone methylation so low in JET cells (Fig. 3)?

3) The data in Fig. 5 suggest that CTCF binding to the proviral genome suppresses Tax expression. However, what is the molecular basis of the increased plus strand viral gene expression in these particular JET cell clones? Is it due to epigenetic changes? Is there increased H3K4me3 marks at the 5’ LTR? Do viral integration sites play a role? Why is HBZ expression unaltered in the different clones?

4) PMA/Ionomycin is not a physiologically relevant approach to induce viral gene expression (Fig. 5). Crosslinking of anti-CD3 and anti-CD28 with agonistic antibodies more accurately mimics T cell activation. Recent studies (Kulkarni et al. 2017 Cell Chem. Biol. 24: 1377-87; Kulkarni et al. 2018 JCI Insight 3: e123196; Mahgoub et al. 2018 PNAS 115: E1269-E1278) have identified stress stimuli as transcriptional activators of the HTLV-1 plus strand. Do stress stimuli (e.g. hypoxia or oxidative stress) also reactivate HTLV-1 gene expression in the latent JET cell clones?

Reviewer #2: Some issues:

(ll 226 ff.):It would have been nice to have looked at one more silencing mark, which is H3K9me3.

ll 345 ff: The discussion of the border phenomenon is important, but should really start out with an explanation of the fact that we do not yet know the point of origin of the methylation area, and that this would be important to identify. Then the discussion of where it might be is more clear. (Also we need some statement of the premise that it can spread.) I don’t think the data really rule out the idea that the 5’LTR might be the source of the methylation domain, though Taniguchi may be right.

ll 244: While the DNA methylation looks like a border issue, the histone modification does not. The summary statement that “loss of the CTCF binding site in HTLV-1 results in increased histone methylation 5’ and 3’ of the CTCF site, suggesting that CTCF binding may act as a boundary element to epigenetic modification” does not quite make sense. This does not seem like a boundary element. Since the effect is the same on both sides, it just seems like a site working in both directions.

It would be nice to have more discussion of the magnitude of the various effects. For example, Fig 2 is impressive. But Fig 5 (expression) is showing generally more modest effects (2-fold or so, often). No point hiding this.

The results of the Bangham paper probably need some more discussion, including how the setting is different than here.

Reviewer #3: No additonal experiments required, but Fig S10 should be presented in the main manuscript and not in the supplement.

**Part III – Minor Issues: Editorial and Data Presentation Modifications**

Reviewer #1: 1) JET cells should be defined when first introduced in the Results section. It is not common knowledge that these cells express tdTomato controlled by Tax responsive elements.

Reviewer #2: Small points:

The reportage of the changes in CpG methylation up- and down-stream of the CTCF BS in the text is someone long-winded and unnecessarily detailed. The numbers are appropriately in the table, and a more general description would be better. Summary (ll. 180 area) is good.

And the format of the fold changes given as “2-31 fold” is somewhat confusing – maybe it needs to be written as “in a range of 2-fold to 31-fold” or something similar. But we don’t really need all the numbers in the text.

(ll. 192 ff) Scoring the 5’LTR vs. 3’LTR for methylation needs explanation – if the assay is by PCR after bisulfite, we need to know how the reads are assigned to either one or the other LTR. (Maybe the reads are long enough to discriminate?)

(ll 199 ff) We need a note as to how strand-specific methylation is read out – presumably seeing C-to-T changes on the appropriate strand (i.e. CpG to TpG as opposed to seeing CpG to CpA change).

(ll 240 ff) It needs to be stated here that (very curiously) both repressive and activating histone marks are increased In the mutant. This is hard to interpret, but it should be laid out here.

Reviewer #3: Line 127: Electrophoretic mobility shift data is not shown. Either show it in the supplement or do not mention it.

Line 174: please cite studies that „commonly used“ JET cells.

Line 238: typo, HTLV-1A should read HTLV-1

Line 254: please provide a reference for the Gini index.

Figures:

All Figures should be formatted according to the journal guidelines. Some figures are too wide.

Fig.2: The left and the right part of the Figure should be adjusted to each other, or the Figure should be arranged in upright format. It would be helpful for the reader if the authors create subpanels, e.g. A) percentage of methylation, and B) change of methylation.

Fig.4A: Why did the authors not analyze HTLV-1 wildtype? Since integration sites of HTLV-1-Delta-CTCF were analyzed, the legend should also read „Delta-CTCF“ instead of CTCF.

Fig. 4B: It is not clear why the authors show the inset. Labeling of the axes is lacking in the inset. The authors state FOXP3 in the legend, but not in the text.

Fig S3: The labeling of the luciferase assay and the western blot should be the same (WT=p12wt, Delta-CTCF=p12tr), otherwise, it is confusing for the reader.

Fig. S4: Copy numbers of HBZ are very low (0.008 per million GAPDH). Could the authors please comment if these low copy numbers are typical for HBZ expression analysis of proviral clones?

Fig. S5: The labeling of subpanels in the legend is mixed up. It should read „in (A) 5’LTR, (D) 3’LTR and HBZ“ and „in (B) PBMCs and (C) JET“; Typo: THLV-1; * and ** should also be explained in the legend

Fig. S7: Why is #15 shown twice? (in Fig. 5B and in FigS7)

Line 549: Typo „).“

PLOS authors have the option to publish the peer review history of their article (what does this mean?). If published, this will include your full peer review and any attached files.

Reviewer #1: No

Reviewer #2: No

Reviewer #3: No
---

## [Editor Report · Decision Letter 1]

22 Apr 2021

Dear Dr. Ratner,

We are pleased to inform you that your manuscript 'Epigenomic Regulation of Human T-Cell Leukemia Virus by Chromatin-Insulator CTCF' has been provisionally accepted for publication in PLOS Pathogens.

Best regards,

Bryan R. Cullen

Associate Editor

PLOS Pathogens

Richard Koup

Section Editor

PLOS Pathogens

Kasturi Haldar

Editor-in-Chief

PLOS Pathogens

orcid.org/0000-0001-5065-158X

Michael Malim

Editor-in-Chief

PLOS Pathogens

orcid.org/0000-0002-7699-2064
---

## [Editor Report · Acceptance letter]

12 May 2021

Dear Dr. Ratner,

We are delighted to inform you that your manuscript, "Epigenomic Regulation of Human T-Cell Leukemia Virus by Chromatin-Insulator CTCF," has been formally accepted for publication in PLOS Pathogens.

Best regards,

Kasturi Haldar

Editor-in-Chief

PLOS Pathogens

orcid.org/0000-0001-5065-158X

Michael Malim

Editor-in-Chief

PLOS Pathogens

orcid.org/0000-0002-7699-2064